# A behavioral and modeling study of control algorithms underlying the translational optomotor response in larval zebrafish with implications for neural circuit function

John G. Holman[1]*, Winnie W. K. Lai[1], Paul Pichler[2], Daniel Saska[1], Leon Lagnado[3], Christopher L. Buckley[1]*

**1** School of Engineering and Informatics, University of Sussex, Brighton, United Kingdom, **2** Department of Neurobiology, University of Vienna, Vienna, Austria, **3** School of Life Sciences, University of Sussex, Brighton, United Kingdom

\* john.g.holman@gmail.com (JGH); c.l.buckley@sussex.ac.uk (CLB)

## Abstract

The optomotor response (OMR) is central to the locomotory behavior in diverse animal species including insects, fish and mammals. Furthermore, the study of the OMR in larval zebrafish has become a key model system for investigating the neural basis of sensorimotor control. However, a comprehensive understanding of the underlying control algorithms is still outstanding. In fish it is often assumed that the OMR, by reducing average optic flow across the retina, serves to stabilize position with respect to the ground. Yet the degree to which this is achieved, and how it could emerge from the intermittent burst dynamics of larval zebrafish swimming, are unclear. Here, we combine detailed computational modeling with a new approach to free-swimming experiments in which we control the amount of visual feedback produced by a given motor effort by varying the height of the larva above a moving grid stimulus. We develop an account of underlying feedback control mechanisms that describes both the bout initiation process and the control of swim speed during bouts. We observe that the degree to which fish stabilize their position is only partial and height-dependent, raising questions about its function. We find the relative speed profile during bouts follows a fixed temporal pattern independent of absolute bout speed, suggesting that bout speed and bout termination are not separately controlled. We also find that the reverse optic flow, experienced when the fish is swimming faster than the stimulus, plays a minimal role in control of the OMR despite carrying most of the sensory information about self-movement. These results shed new light on the underlying dynamics of the OMR in larval zebrafish and will be crucial for future work aimed at identifying the neural basis of this behavior.

## Author summary

In many animals vision is central to the control of locomotory behaviors. In particular, innate motor responses to optic flow allow flying animals to react to gusts of air and fish

**Data Availability Statement:** All data and code for experimental data analysis, model simulation and

fitting are available from a GitHub repository at https://github.com/johnholman/omr-algorithms.

**Funding:** CLB and LL were supported by the Biotechnology and Biological Sciences Research Council (BBRSC) grant BB/P022197/1. Funder website https://bbsrc.ukri.org/ The funders had no role in study design, data collection and analysis, decision to publish, or preparation of the manuscript.

**Competing interests:** The authors have declared that no competing interests exist.

to changes of current. As fish are washed downstream, for example, movement of the riverbed image across the retina evokes forward movement in bouts (short periods of swimming followed by rest periods). It is typically assumed that this translational optomotor response stabilizes the fish's position to prevent it drifting downstream. In larval zebrafish, this response has become a key model system for investigating the neural basis of sensorimotor behaviors in a vertebrate. Here, we combine behavioral experiments and computational modeling to elucidate the underlying control algorithms and explore the detailed relationship between visual stimuli and the initiation and speed of swim bouts. Surprisingly we find that the degree of stabilization is only partial and varies systematically with height above ground, raising questions of the function of this response. We also describe two separate processes underlying bout initiation and bout speed respectively which jointly prove sufficient to predict mean swim speed and degrees of regulation. These findings shed new light on the underlying dynamics which will be crucial for future work to identify the neural basis of this behavior.

## Introduction

The optomotor response (OMR) plays a central role in locomotor behavior in animal species as diverse as insects [1,2], fish [3,4] and mammals [5] including humans [6]. In zebrafish larvae it is one of the earliest and most robust behaviors to develop [7]. The small size and translucency of the larval zebrafish makes it possible to carry out *in-vivo* whole-brain imaging studies [7,8], and the study of the OMR in this species has recently become a key model system for investigating the neural basis of sensorimotor control in a vertebrate [9–12]. However, in comparison with insects, many aspects of the OMR in fish—the behavior itself, its function and the control algorithms underlying it—have received less attention [13]. As Kist and Portugues point out [14] ". . . the OMR, a paradigm that has been extensively used, is still not fully understood, and its comprehensive characterization will undoubtedly reveal further insights into the neuronal circuitry underlying behavior". Indeed, as Krakauer et al. argue [15], in most cases the neural basis of a behavior cannot be properly characterized without detailed prior understanding of the behavior itself and its underlying algorithms.

The OMR in young zebrafish larvae is a relatively complex behavior with two major components [7,13,16]. The usual account is that the fish first responds to rotational components of the whole-field optic flow by turning to face into the water stream. Second, they respond to translational components of the optic flow, specifically rostral/caudal movements across the retina of the image of the ground below, by swimming forward in a series of swim bouts [17] thereby counteracting the movement of its body relative to the ground caused by drifting backwards with the stream. The work reported here focuses exclusively on this second component, the *translational OMR*.

The translational OMR is sometimes characterized as a locomotor behavior that acts in a similar way to the optokinetic reflex (OKR), i.e., by trying to bring the optic flow to zero, it stabilizes the image on the retina, and thus also the position of the animal relative to the environment [12]. However, far from stabilizing the image, intermittent bout swimming during the translational component of the OMR causes the image to oscillate, moving rapidly forwards across the top of the retina during the relatively brief swim bursts and backwards more slowly when the fish is at rest between bouts. In this respect it is quite unlike the OKR which, at least during the slow phase, does stabilize the visual image [18]. The "entrenched belief" [13] that the function of the OMR is to stabilize position relative to the ground has also been challenged,

and indeed it remains unclear whether the degree of regulation it achieves is sufficient in general to prevent drift.

Translational whole-field optic flow, defined as the speed of movement of the ground image across the retina, is certainly a key stimulus for the translational OMR, but other factors may also play a role. For example, forward moving binary or steep dark-light transitions passing locally below the larva's head also promote triggering of swim bouts [14] and there is sensitivity to the spatial frequency of a moving grid stimulus presented below [19]. Here, as the focus is on the role of translational whole-field optic flow, we maximize the importance of optic flow relative to other stimulus attributes by using sinusoidal gratings that lack steep luminance gradients and by scaling the grid spatial period with height to keep the spatial frequency of its retinal image as constant as possible.

Translational optic flow caused by linear movement relative to the ground is inversely proportional to the height of the eye above the ground [20], Fig 1A. For flying insects, the importance of this factor has long been recognized. For example, maintenance of a near-constant optic flow by the translational OMR enables Drosophila to maintain a steady groundspeed that reduces steadily with height despite the perturbing effects of air currents [21] and honeybees to slow down and achieve a grazing landing as they approach the ground [22]. For fish, changes in height while swimming are equally likely to be important but have received relatively little explicit attention. Here we propose that although fish, like insects, may not sense height above the ground directly it nonetheless exerts a profound effect through its impact on the primary effective stimulus, the translational optic flow. The overall optical flow is the sum of the *baseline optic flow* caused by grid movement and the optic flow induced by swimming, both of which are inversely proportional to height (Fig 1B).

Several head-fixed imaging experiments investigating the OMR [9,10,23] have focussed on the effect of *feedback gain*—the extent to which a change in motor effort causes a change in the optic flow [9]–which in that setting is manipulated by changing the relationship between some measure of motor effort and the resulting instantaneous stimulus speed while fixing the actual swim speed at zero. A fish in a free-swimming experiment similarly receives closed-loop visual feedback because changing its speed of swimming through the water changes the speed that the image of the moving grid below passes across the retina (Fig 1C). In this setting we can take the actual swim speed itself as a natural measure of motor effort, in which case the feedback gain is simply the negative reciprocal of the height and can be manipulated directly by changing the height variable (Fig 1C).

Although other factors such as fatigue, viscosity or temperature may cause the swim speed that results from a given motor effort to vary [10,23], for zebrafish larvae swimming in their natural habitat of shallow slowly moving streams in India it seems likely that changes in feedback gain are dominated by changes in height, e.g. halving when a fish rises to a position twice as far from the bottom, and that these changes will take place over much shorter timescales.

Optical imaging of neural activity normally requires a fixed-head procedure and measurement of a variable such as motor nerve activity [10,23] or tail movement [9] that is only correlated with swim speed; these variables do not provide a direct measure of swim speed and therefore of the absolute degree of regulation that would be achieved by the OMR if the fish were free to move. Free-swimming studies, like the head-fixed imaging studies, use a moving stimulus projected below the fish [24] to simulate the effect of a water current, but do allow measurement of the actual swim speed.

In at least one such free-swimming study [25], Severi *et al* observed a close match between stimulus speed and mean swim speed consistent with good regulation for stimulus speeds of up to about 15–20 mm/s. However, this experiment did not explore whether such a match is maintained over a range of different heights. They also found two distinct bout classes with

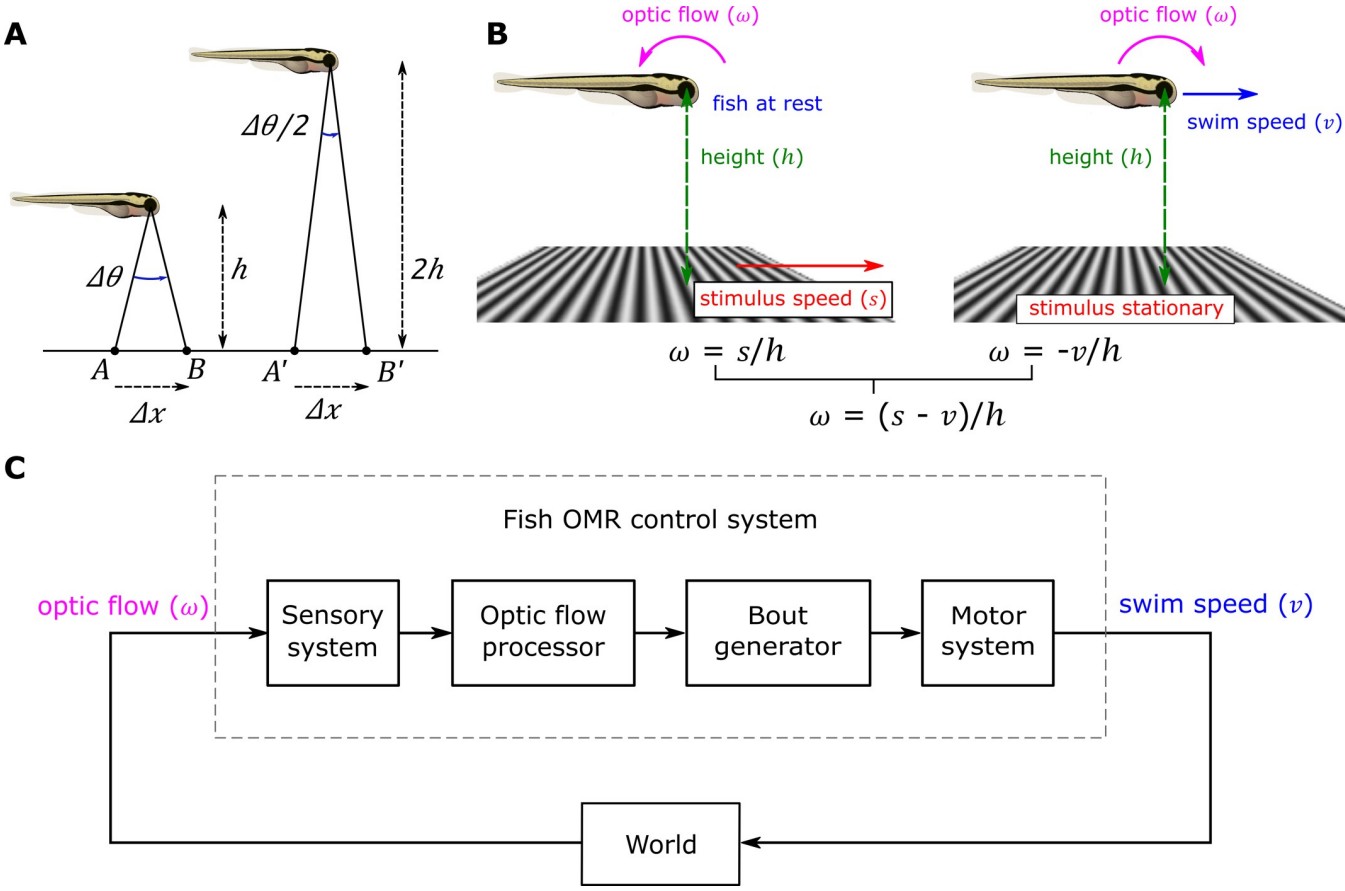

**Fig 1. Optic flow and feedback control of the OMR.** (A) Optic flow is inversely proportional to height. The fish on the left is at rest a distance above a moving stimulus. A point on the stimulus moves forward a small distance $\Delta x$ from A to B over time $\Delta t$, causing the angle of the ray passing from that point through the optical center of the eye to change by $\Delta\theta$ with corresponding optic flow of $\Delta\theta/\Delta t$ radians/s. The fish on the right at twice a height experiences half the change in angle, $\Delta\theta/2$, for the same small stimulus movement in the same time, and thus half the optic flow. If the stimulus speed is $s$, $\Delta\theta = \Delta\theta/h = s\Delta t/h$, so the optical flow $\omega = \Delta\theta/\Delta t = s/h$. (B) Optic flow is a combination of baseline flow and swim-induced flow. As in (A), a fish at rest at height $h$ with the stimulus below moving at speed $s$ experiences a positive (anticlockwise) *baseline optic flow* of $s/h$ rad/s when expressed as an angular velocity (left). Conversely, swimming at speed $v$ over a stationary grid induces a negative optic flow of $-v/h$ rad/s (right). When both fish and the grid are moving the opposing optic flows sum to a total of $(s - v)/h$ rad/s. (C) The overall system is a closed-loop feedback system with gain inversely proportional to height. Any change in the swim speed output causes a change in the optic flow input, which in turn feeds back to affect future swimming. Following [9], the system's *feedback gain* can be defined as the change in optic flow caused by a small change in the swim speed taken as an index of motor effort, divided by that swim speed change, i.e., the derivative of the input with respect to the output $d\omega/dv$. The optic flow $\omega = (s - v)/h$, with $s$ and $h$ constant for any given trial, so the feedback gain $d\omega/dv = -1/h$. Like the optic flow itself, the feedback gain is inversely proportional to the height of the fish above the ground–in fact it is just the negative reciprocal of the height.

different kinematic characteristics: slow bouts and fast bouts. Slow bouts predominated for moderate stimulus speeds, while fast bouts started to emerge at about 12.5 mm/s and increased in proportion thereafter. When considering degrees of regulation, we focus here on stimulus speeds of up to 12 mm/s only and in general our conclusions should be taken to apply to slow bouts only.

Markov et al [12] proposed perhaps the most detailed and complete account to date of the translational OMR in larval zebrafish. It includes a model of a feedback controller that generates the behavior itself and proposes an additional mechanism that modulates parameters of that controller over longer timescales than those considered here to adapt to environmental changes. However, in their model the swim speed for all bouts is fixed at 20 mm/s and the feedback controller acts only to initiate and terminate bouts. In reality, mean swim speeds during bouts are observed to vary with stimulus speed [25]. This is a prominent feature of the OMR

and a complete account must also propose algorithms for the control of swimming speed during bouts.

In this paper, we address some of the gaps in understanding through free-swimming experiments that manipulate feedback gain through changes in the height of the fish above a moving stimulus below, allowing absolute measurements of the degree of regulation achieved, and through detailed computational modeling. Together these lead to several new conclusions about the OMR and its underlying control algorithms. The experiments show that the degree of regulation achieved by the translational OMR is only partial and varies substantially, and systematically, with height. Although it certainly reduces drift, any assumption that the OMR succeeds in maintaining position seems too strong. We also observe that the relative speed profile of individual swim bouts takes an invariant form that is independent of absolute overall bout speed. This suggests that although bout initiation is separately and explicitly controlled, bout speed, termination and active duration may vary together and share a second common control process.

Next, we present a series of algorithmic-level models of the OMR, based on the experimental findings and drawing initial inspiration from control theory, that propose mechanisms for the control of bout speed as well as bout initiation. First, we describe a single factor model that proposes that the same process of sensory integration underlies both bout initiation and the control of bout speed. This model proves sufficient to explain overall mean swim speeds but not the observed bout patterns. We then describe dual factor models in which optic flow is processed differently for bout initiation and speed control and find that such a model can predict the bout patterns observed as well as mean swim speeds. We conclude that the control of bout initiation and bout speed are at least partially decoupled: the control of bout speed involves leaky integration of optic flow while bout initiation appears to depend on the immediate unintegrated optic flow perhaps in conjunction with inhibitory motor integration. Furthermore, the final dual factor model suggests that although the rostral translational flow induced by swimming carries much of the available sensory information about self-movement, surprisingly it may play little or no part in the control of either bout initiation or bout speed and therefore of the translational OMR. Finally, we discuss some of the implications of these new findings.

## Results

### Free swimming experimental setup

We built a free-swimming behavioral rig (Fig 2A) which mimics the situation where the larval zebrafish swim at various heights against a water stream by displaying visual motion below to induce translational optic flow. We tracked the 2D position of individual fish from above in separate longitudinal acrylic channels of width 10mm and depth 8mm where sufficient length was provided for each of them to reach steady-state swimming for at least 90% of the trial distance. Moving sinusoidal gratings, scaled up in proportion to height to keep the spatial frequency of the retinal image as constant as possible and close to an optimal value [26], were presented on a computer monitor from below. One independent variable, height, was varied between three levels (low, medium, high) by setting the vertical distance between the screen and the midpoint of the channel to be 8, 32 and 56 mm respectively. The spatial frequency of the grating, 0.0156 cycles per degree (cpd) at the center of the channel, was that found to be optimal at the medium height (S10A Fig). It was subsequently confirmed to be optimal also for both the low and high heights (S10B Fig). This suggests that although the perceived spatial frequency would be somewhat greater than 0.0156 cpd due to refractive effects [27] with some variation between height conditions, this variation had little effect here (see S11 Fig and Methods for further details).

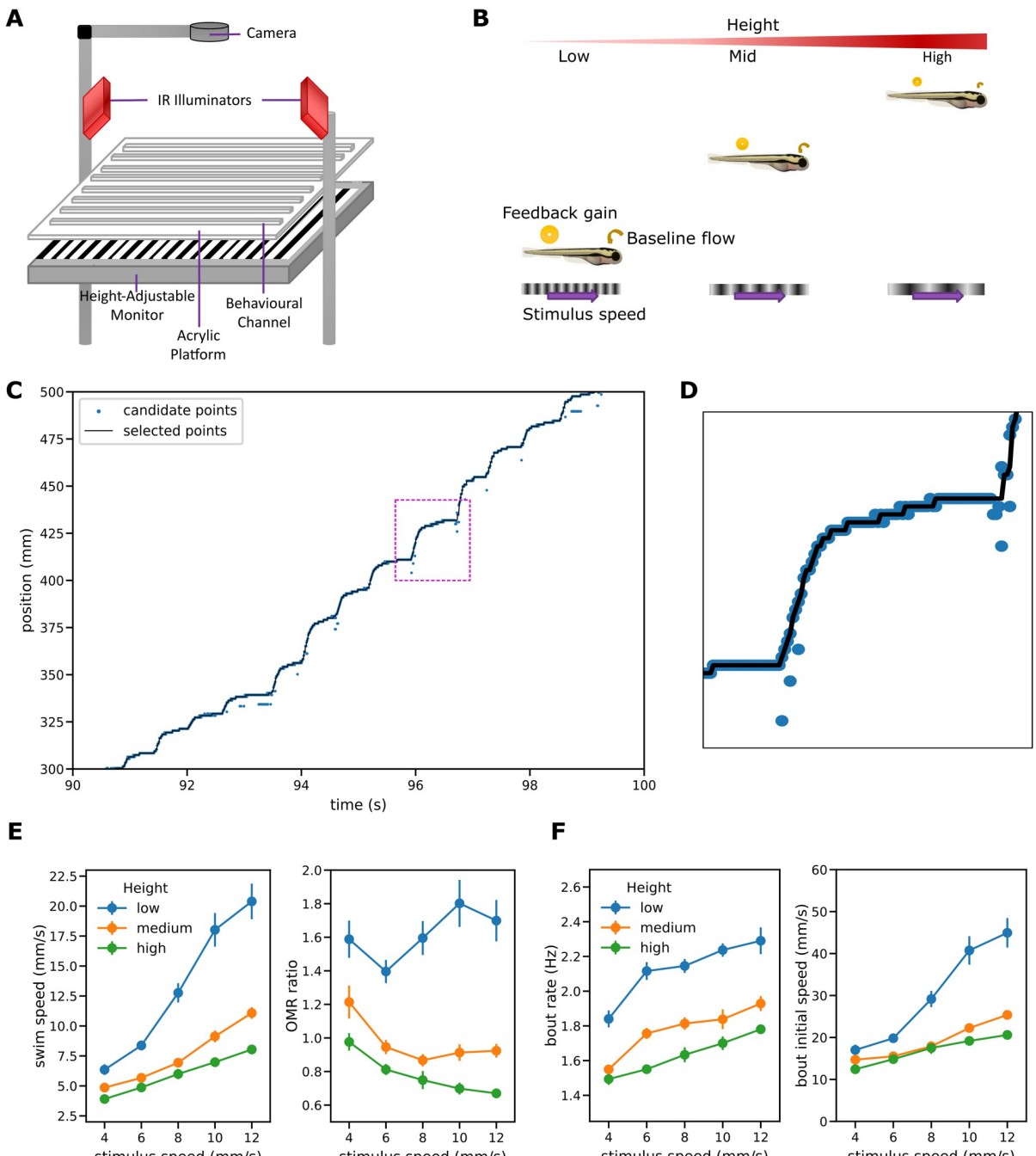

**Fig 2. Free swimming setup and the OMR regulation experiment.** (A) Schematic illustration of the experimental apparatus. (B) OMR regulation procedure. Each fish is tested at one of three different heights and the same set of stimulus speeds. For a given stimulus speed, baseline optic flow–the optic flow experienced when the fish is at rest–and feedback gain both decrease as the height increases. (C) Sample trace showing position of the fish from the start of the channel. The light blue points show possible positions for the center of mass at each time step; the black line shows the positions selected as most probable by the tracking algorithm from those candidates. Dotted rectangle shows region shown in (D). (D) Magnified portion of trace containing a single bout. (E) Mean swim speed and OMR ratio for the OMR regulation procedure. OMR ratio is defined as the ratio of the mean swim speed observed during OMR trajectories to the speed of the grid stimulus. Error bars here and elsewhere: standard error. (F) Bout patterns for the OMR regulation procedure.

The other independent variable, stimulus speed, was controlled via a custom-written program. We took a high-throughput approach and recorded batches of eight fish simultaneously, each in a separate channel. Using a mixed design to reduce disengagement from fatigue, each batch was subjected to only one of the height conditions but all the testing speed conditions. Each session included two *trial traverses* lasting 60s with the grating moving forward during the first traverse and moving in the opposite direction for the second.

To investigate the OMR in isolation from other behaviors such as exploratory swimming as far as possible, we focused on analysis of behavior during *OMR trajectories*: periods of the trial traverses during which the fish swam with a mean speed of at least 0.5 mm/s and in the same direction as the stimulus and thus behaved in a manner at least consistent with the OMR (see Methods for further details). An *OMR-consistent* bout was defined as a swim bout occurring during an OMR trajectory. The *mean swim speed* for a session was defined as the mean of the mean swim speeds for its trial traverses, giving the two trials equal weight; the mean swim speed for a trial traverse was measured as the total change in position over all its OMR trajectories divided by the total duration of those trajectories.

To investigate the bout structure underlying the mean swim speeds, we analyzed the position data to identify *bout initiation* events and thus calculate the mean *bout rate* for OMR-consistent bouts. We also calculated measures related to the speed or intensity of each bout: *displacement* defined as the distance travelled between the start of that bout and the start of the following one, *initial bout speed* defined as the mean swim speed over the first 100 ms of the bout, and *peak bout speed*. We chose initial bout speed as the primary measure of bout strength as this is less noisy than the peak instantaneous speed and unlike displacement is not confounded with the timing of the following bout. Note that we did not descend another level to capture and analyze properties such as tail beat frequency or amplitude of the detailed swimming movements that underlie and are correlated with bout speed [25]. The focus here is on central control mechanisms rather than details of sensory or motor processes, and in that context it is the kinematic *consequences* of the swimming movements–the speeds through the water caused by the movements–rather than details of the movements themselves that are most relevant. In the free swimming setting those kinematic consequences can be measured directly.

## OMR regulation

During the translational component of the OMR, fish swim while maintaining an orientation that faces into the water stream thus reducing their speed over the ground. To examine the degree to which regulation is maintained despite the changes to optic flow induced by changes in height, and thus throw light on the underlying control mechanisms, we compared the swim speeds and degrees of regulation using the basic *OMR regulation procedure* in which different groups of fish swam at different heights with the grid below moving at the same set of fixed speeds (Fig 2B). *OMR ratio*, defined as the ratio of mean swim speed to stimulus speed, served as the measure of regulation; perfect regulation corresponds to an OMR ratio of 1, under-compensation to a ratio less than 1 and over-compensation to a ratio of more than 1.

## The degree of regulation varies systematically with height

Most swimming bouts during the trial traverses (83% overall) were in the same direction as the moving grid confirming elicitation of the OMR. This proportion did not vary systematically with stimulus speed (S1 Fig.) suggesting that even the slowest stimulus speed of 4 mm/s was fully capable of engaging OMR behavior. Within the extracted OMR trajectories, which made up on average 73% of the full trial traverse period, 99.6% of bouts were in the same direction as the stimulus.

Mean swim speeds during OMR trajectories increased with stimulus speeds (Fig 2E). This is consistent with a regulatory effect, but the degree of regulation varied with height. Welch F tests showed a significant effect of height on both mean swim speed ($F_{(2,56.1)} = 67.4$, $p<0.001$) and OMR ratio ($F_{(2,57.2)} = 70.9$, $p<0.001$). At the intermediate height, fish swam at a similar speed to the stimulus. However, regulation at other heights was far from perfect: during OMR trajectories fish swimming at low heights overcompensated, swimming faster than the stimulus, while at a higher level tended to undercompensate. The overall mean OMR ratio for the intermediate level group, 0.97, did not differ significantly from the value of 1 associated with perfect regulation ($t(31) = -0.62$, $p = 0.54$). However, the mean OMR ratio for the low level group, 1.62, was greater than 1 ($t(31) = 9.67$, $p<0.001$); that for the high level group, 0.78, was less than 1 ($t(31) = -7.57$, $p<0.001$).

The increases in swim speed as height reduced resulted from increases in both bout rate and bout speed (Fig 2F). Bout rate was defined as the frequency of bouts occurring during OMR swim trajectories; bout intensity as the initial bout speed (the mean speed over the first 100ms). Welch F tests showed a significant effect of height on both overall mean bout rate ($F_{(2,60.8)} = 82.2$, $p<0.001$) and overall mean bout initial speed ($F_{(2,57.8)} = 35.1$, $p<0.001$)

To assess the overall OMR ratio when periods of OMR-consistent behavior alternate with periods when other behaviors may dominate, we also analyzed the full period of the trial traverses up to the time when the end of the channel was first reached without extracting OMR trajectories (S2 Fig). The pattern observed was generally similar to those for OMR trajectories but with lower ratios for all conditions as expected. The overall OMR ratios again decreased systematically with height ($F_{(2,56.3)} = 23.0$, $p<0.001$) but were now significantly less than 1 for both medium ($t(31) = -6.3$, $p<0.001$) and high height conditions ($t(31) = -15.4$, $p<0.001$) while the mean OMR ratio for the low height group did not differ significantly from 1 ($t(31) = 0.95$, $p = 0.35$)

## Baseline optic flow procedure

*Baseline optic flow* can be defined as the optic flow experienced when the fish is not swimming, induced in the natural situation as the water stream sweeps the fish backward and in free swimming experimental procedures by the fixed forward movement of the grid stimulus. As feedback gain and baseline optic flow are both inversely proportional to height, the effects of each are confounded in the OMR regulation procedure.

In a free swimming setting the baseline flow can be held constant by increasing the stimulus speed in proportion to height. The resulting *baseline optic flow procedure* (Fig 3A) enables feedback gain to be manipulated by changing the height while holding other factors constant, enabling investigation of the effects of behavior-induced sensory changes in an otherwise unchanged sensory environment.

This procedure is analogous to that of head-fixed virtual reality experiments in which feedback gain is manipulated by changing the relationship between swimming effort and stimulus speed while holding constant baseline stimulus speed and height, and therefore also baseline flow. Such experiments have shown that a variable correlated with swim speed such as the amplitude of tail movements [9] or motor nerve activity (2) typically increases as gain decreases even though at time points when swim bouts are initiated the environment remains constant.

Employing this baseline flow procedure, we investigated the effect of varying baseline flow from 0.1 rad/s to 0.5 rad/s at the same three heights as before. The corresponding stimulus speeds ranged from 0.8 mm/s to 28.0 mm/s (Table 1).

We found that mean swim speeds increased with height even when the baseline flow is held constant (Fig 3C), as expected from the virtual reality results [9,28]. The sole exception was the

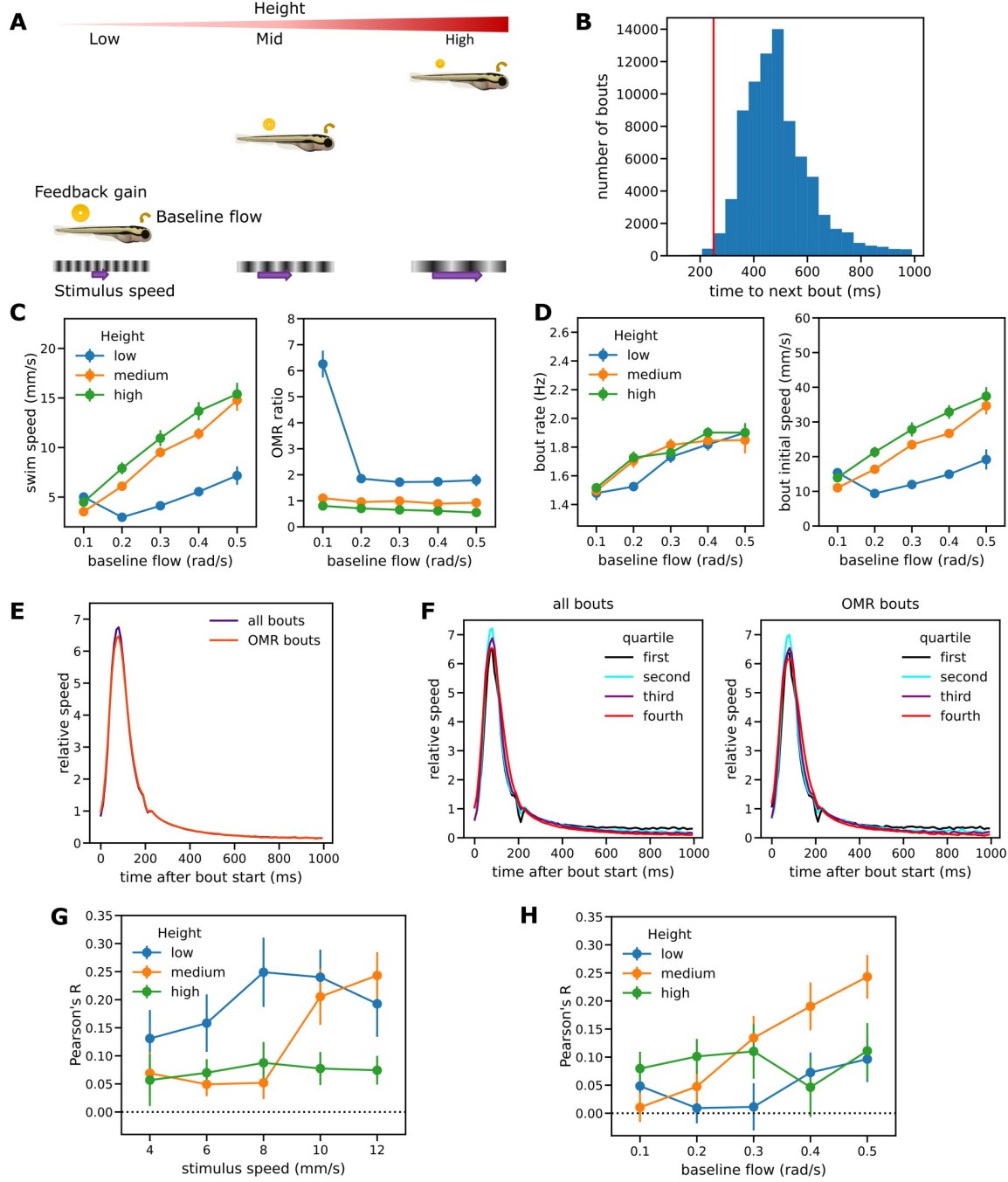

**Fig 3. Experimental results for the baseline flow experiment and bout analysis.** (A) Baseline optical flow procedure. Each fish is tested at one of three different heights with the stimulus speed increased in proportion to height. Feedback gain decreases as the height increases but the baseline optical flow remains constant. (B) Histogram of intervals from the start of one bout to the start of the next for OMR-consistent bouts pooled from both procedures. The interval exceeded 250ms for 99.5% of bouts C) Mean swim speed and OMR ratio for the baseline flow procedure. (D) Bout patterns for the baseline flow procedure. Bout speed increased with height, but bout rate did not. (E) Overall relative bout speed profile from both experiments for all observed bouts and for OMR-consistent bouts only. The profile was not sensitive to the behavioral context. (F) Separate relative speed profiles for bouts split into quartiles by initial bout speed (swim speed averaged over the first 100ms of a bout). The relative speed profiles were essentially independent of the absolute bout speeds and again similar for all bouts (left) and OMR-consistent bouts (right). (G-H) Bout speed and time to next bout were positively correlated for OMR-consistent bouts in both the OMR regulation (G) and baseline flow procedures (H).

**Table 1. Stimulus speeds (mm/s) for the baseline flow experiment.**

| | Baseline flow (radians/s) | | | | |
|---|---|---|---|---|---|
| | **0.1** | **0.2** | **0.3** | **0.4** | **0.5** |
| low height | 0.8 | 1.6 | 2.4 | 3.2 | 4.0 |
| medium height | 3.2 | 6.4 | 9.6 | 12.8 | 16.0 |
| high height | 5.6 | 11.2 | 16.8 | 22.4 | 28.0 |

condition with the lowest stimulus speed (0.8mm/s) which resulted in a very high OMR ratio of 6.26. This outlying data point was excluded from all further analyses as it seems likely that the very slow stimulus speed was insufficient to elicit OMR behavior, allowing spontaneous swimming to dominate. As in the OMR regulation procedure, Welch F tests showed a significant effect of height on both mean swim speed ($F_{(2,59.2)} = 50.9$, $p < 0.001$) and OMR ratio ($F_{(2,55.7)} = 59.5$, $p < 0.001$).

To achieve full regulation in this experiment swim speeds for a given baseline optic flow must increase in proportion to height. The actual height above the stimulus for a given height condition could vary by up to 8mm (the height of the swimming channel), but the ratio of heights between the high- and low-level conditions was always at least 4.33. The largest observed ratio of swim speeds was only 2.67, showing again that the OMR achieves only partial regulation.

At times immediately prior to bout initiation the observed swim speeds were almost always very close to zero and thus optic flows close to the baseline level, suggesting that a reflex-like control mechanism based exclusively on the immediate optic flow at the point of bout initiation may not be sufficient. Instead, as others have argued [9,10], sensory integration is likely to be involved. However, because of the significant sensory delays associated with visual processing, when two bouts occur in quick succession a fish may still be experiencing some of the sensory consequences of the first bout, namely a reduction in the optic flow below the baseline level, at the time it initiates the second. The finding that swim speeds are sensitive to feedback gain as well as to baseline stimulus speed is therefore not sufficient to rule out a simple mechanism without further analysis.

## Bout speed profile has a fixed form

To investigate the bout patterns underlying observed mean swim speeds we identified the times at which each bout was initiated. Fig 3E shows the profile of instantaneous swim speeds for times after the start of a bout averaged over all bouts observed during the test traverses, and normalized so that profile speeds are relative to the mean bout speed taken over the first 1s (see Methods for details). A very similar profile was observed when only OMR-consistent bouts were included (Fig 3E). The observed profiles also demonstrate that swim speed continues to decline from a low baseline for some time after active swimming has ceased; this is likely to be due to inertia.

Bouts were then split into quartiles by initial bout speed and separate profiles calculated for each quartile. These profiles were very similar to each other, whether considering all bouts or OMR-consistent bouts only (Fig 3F), suggesting that the relative bout speed profile takes a fixed form that is insensitive both to the absolute bout speed and to the behavioral context (whether bouts occur as part of the OMR or another behavior such as exploration). For weak and strong bouts alike, swim speed reached a peak of about 6.75 times the mean bout speed about 80 ms after bout initiation and then dropped to fall below the mean speed again about 200 ms after initiation. Moreover, similar profiles were obtained when bout speed was measured by peak speed (S3A Fig) or total displacement (S3B Fig), suggesting that initial speed, peak speed and bout displacement are essentially equivalent as measures of bout speed. We

chose initial bout speed as it is less noisy than peak instantaneous speed and unlike bout displacement is not confounded with the occurrence of a closely following second bout.

More generally, these results suggest that a single variable may be sufficient to capture and explain variations in bout intensity whether manifested by variations in bout speed, duration of active swimming during bouts, or the rate or amplitude of swimming movements. In particular, a fixed relative speed profile implies that bout duration (the duration of active swimming preceding the interbout interval) will increase along with bout speed whenever bout termination is defined by a measure of bout intensity falling below some threshold. For the OMR regulation experiment, with bout termination defined as the point at which swim speed first dropped below 5 mm/s, bout duration indeed increased with stimulus speed in a manner consistent with that observed by Severi et al [25] when identification of bout termination was based on tail movements (S4 Fig).

## Control of bout speed and bout initiation are decoupled

The pattern of bout measures for the baseline flow procedure was qualitatively different from that for the OMR regulation procedure (Fig 3D): changes in height still resulted in changes in bout speed ($F(2,59.8) = 40.5$, $p<0.001$) but bout rates remained essentially unchanged ($F(2,59.9) = 0.15$, $p = 0.86$). Independence of bout rate and height was not an artefact of employing a wide range of stimulus speeds: even when conditions with speeds slower than 3 mm/s (which may not fully engage the OMR) or greater than 15 mm/s (when fast bouts are likely) were excluded from the analysis, for a given baseline flow the bout rate did not depend upon height (S1 and S5A Figs).

The finding that manipulating height affected mean swim speed via changes in both bout rate and speed for the first procedure, but via bout speed alone for the second, suggests that the mechanisms underlying bout initiation and speed are at least partially decoupled.

## Modeling the translational OMR

**Operation of the single process model.**   Simple reflexes such as the tonic stretch reflex have often been modeled as feedback control systems [29]. A similar approach may be taken for control of the OMR by treating optic flow as the feedback signal in a closed loop control system [12]. In our formulation, an *optic flow processor* transforms the *sensed optic flow* to a *control signal* that drives a *bout generator* (Fig 1C). The bout generator in turn outputs a motor signal reflecting the instantaneous swim effort as it varies through the bout, while the motor system is responsible for generating the pattern of rapid tail beats that propel the fish forward through the water. From a control engineering perspective [30] we might view the flow processor as a controller and the combination of bout generator and motor system as the plant to be controlled. The usual control engineering task is to design a controller that achieves desired system behavior given knowledge of the plant; here the modeling task is to reverse engineer both the controller and plant to reproduce observed behavior. In particular we aimed to find the simplest possible model in which the optic flow stimulus is sufficient to generate the behavior we observed, including the dependence on height, without requiring the fish to estimate height directly or to exploit other stimuli such as visual texture or lateral line stimulation.

We start by explaining the operation of the model shown in Fig 4C in which the bout generator has a single input that controls both bout initiation and speed. This *single process model* proved sufficient to predict mean swim speeds and therefore degrees of regulation. As might be expected given the observed decoupling of bout rate and bout initial speed, it turned out to be unable to predict behavior at the level of bout patterns, but it still serves as a good starting point for explaining the evolution of the more capable but more complex dual factor models presented subsequently.

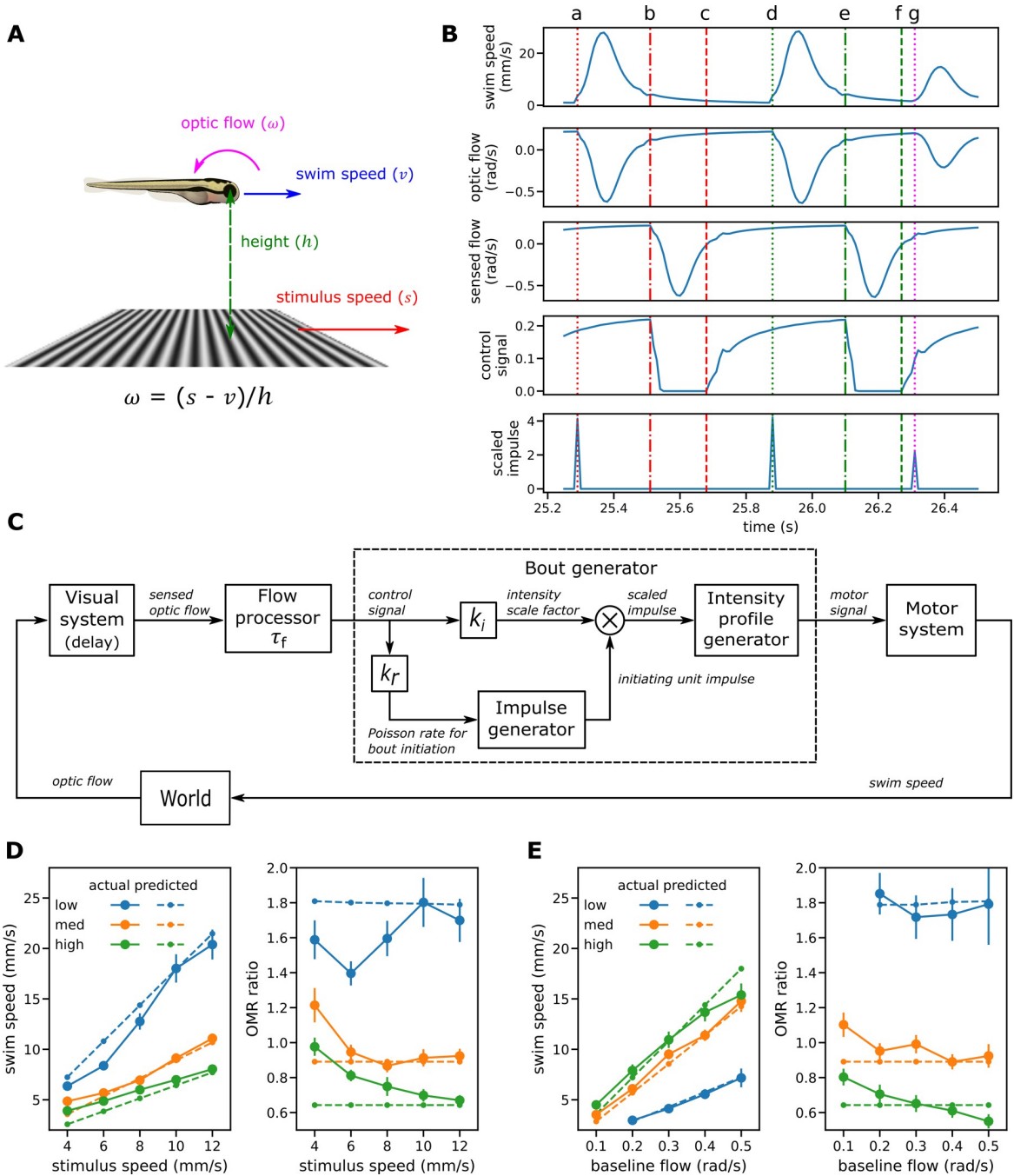

**Fig 4. Modeling I: Single process model.** Operation of the single process model. For further details see main text and the more formal model description section of the Methods. (A) Fish swimming in the same direction as a moving grid below. The resulting translational optic flow, expressed as an angular velocity, is the ratio of the difference between grid speed and swim speed and the height of the fish above the grid, as explained in Fig 1B. (B) Sample traces generated by the single factor model. The top two traces show the observable variables swim speed and optic flow; the bottom traces show three key model variables. The vertical lines show events occurring during the first bout (red, a-c), second bout (green, d-f) and third bout (magenta, g). (C) Block diagram for the single factor model. The sensed flow is a delayed version of the actual optic flow. The flow processor smooths the sensed flow by applying leaky integration and rectifies it to produce the control signal input to the bout generator. Within the bout generator, the amplified control signal drives an impulse generator that implements a Poisson process for the event of bout initiation. The same control signal also modulates the amplitude of the initiating impulses from the impulse generator to produce scaled impulses input to the profile generator. The profile generator in turn generates a swim bout that starts when the scaled impulse is received and whose speed follows the fixed relative swim speed profile observed scaled up by the amplitude of the scaled impulse. The model has three parameters to fit to data: the time constant for leaky integration ($\tau_f$) and the gains applied to the control signal to determine the Poisson rate for bout initiation events ($k_r$) and the amplitude

of scaled impulses ($k_i$). (D-E) Comparison of swim speeds and degrees of regulation as observed in the experiments and predicted by the single factor model when fitted to mean observed swim speeds for the OMR regulation procedure (D) and the baseline flow procedure (E).

When a fish is swimming more slowly than the speed of the grid and in the same direction, the image of the stimulus below moves across top of the retina in a rostral to caudal direction. The magnitude of translational optic flow expressed in radians/second is the ratio of the difference between the stimulus speed and swim speed and the height (Fig 4A). It is positive or "forward" when the fish is swimming more slowly than the stimulus, negative or "reverse" when the fish is swimming more quickly, and zero when the speeds are the same. This is illustrated in the top two traces of Fig 4B showing the operation of the model over a sample period during which it generated three swim bouts. After the first bout is initiated at the time indicated by the red dotted line (a), the swim speed increases and then decreases as shown in the first trace; the second trace shows the corresponding changes in optic flow. As a fish usually swims much more quickly than the grid stimulus during the early part of the bout, typically the flow drops, reverses in direction and becomes strongly negative before returning towards the moderate positive baseline induced by grid movement as the swim speed decays back towards zero.

Fig 4C shows a block diagram of the single-factor model, whose input is the optic flow and output the resulting swim speed, while the lower three traces of Fig 4B show traces for three of this model's key internal variables. We assume that the visual system provides an accurate but delayed estimate of the actual optic flow. This is shown as the *sensed flow* in the third trace; the red dashed-dotted line (b) shows the time at which the modelled fish first senses the decrease in optic flow produced by the swim bout that occurred some 220 ms earlier. The *flow processor* shown in the block diagram then smooths the sensed flow signal by applying leaky integration with a time constant of $\tau_f$ and rectifies it to produce the *control signal* input to the bout generator shown in trace 4 of Fig 4B. The red dashed line (c) shows the time at which this signal becomes positive again following the first bout; from this time onwards, it is possible for another bout to be triggered. A new bout is initiated when the *impulse generator* emits a unit pulse lasting a single timestep. This occurs on any given timestep with a probability proportional to the control signal that generally increases steadily during the interbout period, with $k_r$ the constant of proportionality. Here the initiation event for the second bout occurs at the time indicated by the green dotted line (d).

For the single process model, the speed of swimming during the bout also varies in proportion to the single control signal with $k_i$ the constant of proportionality. As shown in the block diagram, the output of the impulse generator gates the amplified control signal. The result is a *scaled unit impulse* occurring at the start of the bout whose amplitude is proportional to the control signal (trace 5). When a bout is initiated, the *speed profile generator* outputs a motor signal that results in swimming whose speed profile follows the observed relative bout speed profile scaled up in proportion to the magnitude of the scaled impulse. Bout initiation and speed are strongly coupled in the single factor model, as the probability of a bout starting and the speed of the bout are both proportional to the same single control signal. For example, for the third bout in the example, the time from when the control input first turns positive (dashed green line f) to when that bout is triggered (magenta dotted line g) happens to be much shorter than the corresponding period for the second bout, resulting in both a shorter interbout interval and a relatively weak third bout.

**Optic flow processing and leaky integral control.** Having outlined the single factor model's overall operation, we now give the reasoning behind some of the design choices.

Within engineering, most controllers found in practice are variants of proportional-integral-derivative (PID) control, a simple controller design that has been applied successfully to a

vast range of control tasks [30]. The input to such a controller is an error signal representing the difference between the desired and actual values of some observable variable. For proportional (P) control the output is just the immediate input error signal amplified by a gain factor, the *proportional gain*; for integral (I) control it is the integrated error signal amplified by the *integral gain*. For proportional-integral (PI) control a mixture of the two provides both timely response to changes in the error signal by virtue of the immediate proportional component and elimination of steady state errors by virtue of the integral component. For full PID control, the derivative of the error signal makes a further contribution, but this is very sensitive to sensor noise and often not used in practice [30]; it is not considered further here.

When modeling the translational OMR it seems reasonable to identify the error signal directly with the forward-backward translational optic flow: to achieve regulation, if this component of the optic flow is on average positive (forward) the fish should speed up and if negative (reverse) should slow down.

Traditional integral control, whether or not combined with proportional control, in general results in systems that have little or no steady state error. The experimental results above show that this is not the case for the OMR: significant degrees of over-compensation are observed for swimming at low level and under-compensation at high levels. This may not be surprising, as standard integral control requires perfect memory of the integral error while integration in biological systems is often considered to be leaky, as for example in leaky integrate-and-fire neuron models [31,32]. The model for optic flow processing adopted here is therefore a generalization of standard integral control termed here *leaky integral (LI) control* in which the optic flow error signal is accumulated through time but also decays exponentially to zero with a fixed time constant (see Methods for details).

Leaky integral control could be combined with P control to create a generalized PI controller, but to keep the model as simple as possible we start by considering only LI control. However, P only control is included and treated as the limiting case of LI control as the time constant tends to zero.

**Bout generation.**   A key difference between the swimming of zebrafish larvae and the output of most engineered systems is that movement is intermittent. The average swim speed and degree of regulation achieved depend on at least two factors: *bout speed*, the swim speeds achieved during a bout, and *bout rate*.

Relative swim speeds during the bouts observed in these experiments followed a fixed temporal pattern (Fig 3E and 3F) independent of absolute bout speed. This observation suggests that individual swim bouts resemble fixed action patterns whose form is determined intrinsically rather than guided by external stimulation and that are ballistic in nature, i.e., once a bout is launched its speed trajectory follows a fixed course.

Markov et al [12] suggest that optic flow perception is subject to significant sensory delay of the order of hundreds of milliseconds; here we adopt their estimate of 220ms, which is similar to the duration of active swimming in the bouts observed here. As they point out, swimming cannot be affected by its impact on optic flow until after that delay period is over, implying that the initial portion of a bout is necessarily ballistic. The sensory delay rules out accounts of speed profile generation based on visual sensory feedback; indeed, if their estimate of the delay is correct, the short bouts observed here must be ballistic for almost their entire duration as active swimming was almost always less than 250ms.

These observations suggested modeling bout profile generation as a linear time-invariant (LTI) system [33] whose impulse response is the observed normalized speed profile and whose input is a single short duration impulse at the time of bout initiation with magnitude proportional to bout speed. This captures, in a simple way, the key features that bouts are ballistic with their speed following a fixed temporal pattern. One implication of such a model is that

the swim speed trajectory of an individual bout is completely determined by the timing and magnitude of its initiating impulse.

For bout initiation, we followed Portugues et al. [34] in proposing a stochastic process: bouts initiation events follow an inhomogeneous Poisson process whose instantaneous rate depends on the sensed optic flow. Another option is a noisy threshold model, but they found a Poisson process to give a better fit at least when modeling the latency for starting to swim following onset of stimulus movement.

Sensory delays in the visual system of the order of 220ms imply that the sensed optic flow remains high for almost the entire bout, as does the control input to the bout generator. To prevent immediate re-triggering of new bouts we therefore also imposed a *refractory period* during which bout initiation events are prohibited. The refractory period was not treated as a free model parameter but set from the experimental data at 250ms; the period between successive bouts was greater than this for over 99% of the bouts observed (Fig 3B).

**Single factor OMR model can predict mean swim speed and degrees of regulation.** Fig 4C shows the simplest possible way to combine optic flow processing and bout generation as described above to yield a complete model of OMR control (see Methods for mathematical details). A single control signal, the result of leaky integration and rectification of the sensed optic flow, governs both bout speed and bout initiation.

The impulse generator implements the Poisson process and outputs a unit impulse when a bout initiation event occurs. This impulse is multiplied by the intensity scale factor to determine the scaled impulse input to the intensity profile generator. This in turn implements an LTI system whose impulse response is taken directly from the observed relative bout speed profile. The motor signal output here thus reflects directly the swim speeds through the water to be achieved, which are then realized by the motor system.

Note that we did not attempt to model sensory or motor systems or the physics of the environment. A more complete model, for example, would specify the processes of motion detection including contributions from local features [14] and details of how tail beats are generated along with the physics of how the pattern of tail beats translates into propulsive force and how this translates into forward motion. Here however the focus is on the control of swim speed at the bout level, one level of abstraction above the tail beat level; we seek a model that given the environmental conditions of height and stream can predict characteristics of the bout patterns (bout rate and bout speed) from which mean swim speed and therefore regulation emerges.

To fit the models, the data from both experiments were pooled and relative mean prediction errors calculated as described in the Methods section. The parameter set that minimized the overall relative prediction error for the mean swim speed (Table 2, middle column) was then used to generate synthetic data from the model for both the OMR regulation and baseline flow procedures.

When only predicted mean swim speed is considered, optimizing parameters for the single factor model to minimize swim speed prediction error alone gave a fairly good fit to the

**Table 2. Optimal parameters for the single process model.**

| Parameter | Swim speed fit | Bout rate and intensity fit |
|---|---|---|
| $\tau_f$ | 0.071 | No integration |
| $k_i$ | 250.46 | 22.36 |
| $k_r$ | 50092 | 39.76 |

A first set of model parameter values was found by minimizing the swim speed prediction error. A second set was found by simultaneous minimization of the relative bout rate and bout initial speed prediction errors. (See methods section for details)

experimental data for both procedures (Fig 4D and 4E). The main qualitative findings are replicated: in the OMR regulation procedures, fish swim faster than the stimulus when at a low level, at high level tend to swim slower, and at the intermediate height roughly keep pace (though the experimental data shows some deviations from this pattern at slow stimulus swim speeds).

**The single factor OMR model does not predict bout characteristics.** In contrast, the bout patterns generated by the model when fitted to mean swim speed differed greatly from those observed. In itself this is not surprising as the system is underdetermined: a given mean swim speed can be achieved by a high rate of weak bouts or a low rate of strong ones. Refitting the basic model to minimize both bout rate and bout speed errors simultaneously rather than the mean swim speed error resulted in a very different parameter set (Table 2, rightmost column) and an improvement in bout characteristics. However, the fit is still quite poor and predictions for mean swim speed are worse than when fitted directly to swim speed (Fig 5A). In particular, the single factor model when fitted to the bout measures predicted that height has rather little effect on mean bout rates for the OMR regulation procedure but a large effect for the baseline flow procedure; the pattern observed was the opposite (Fig 5C).

It seems then that the single factor model in which the same internal variable controls both bout rate and bout speed can make reasonable predictions about mean swim speed but fails to capture how those speeds are realized at the bout level. This is not surprising: the baseline flow experiment suggested that bout speed varies with height when baseline flow is controlled but bout rate does not, suggesting that the mechanisms responsible for bout initiation and for intensity are decoupled to a greater extent than can be accommodated by a single factor model.

**Improving the bout initiation model.** To investigate whether the bout generator component of the single factor model can reproduce the mean bout rates observed when relieved of the responsibility of also determining bout speed, we set the intensity scale factor for the bout generator to be the mean of the bout speeds observed in the corresponding experimental condition and fitted the resulting model to the experimental data by minimizing the mean bout rate error alone. The relative bout rate prediction error was lower than before, but still substantial and the finding that bout rate is essentially independent of height when baseline flow is controlled still could not be reproduced (S6A Fig). Importantly, when intensity was determined separately the best fit for bout rate was obtained with no integration at all (flow integrator output the same as its input) suggesting that bout initiation unlike bout speed may depend only on the immediate visual stimulus.

What is needed for the model to replicate our observation that bout rate was independent of height when baseline flow was held constant? This is challenging if bout initiation is based on the visual stimulus alone. In the baseline flow experiment, fish swam during bouts more slowly at low level than at high level, but insufficiently so to compensate completely for the height difference. Consequently, the reduction from the same baseline in optic flow was substantially greater for fish swimming at low level than at high level. This might be expected to cause bout rate to reduce with height, as indeed is predicted by the model, but this was not observed in reality. Similarly reductions in height would act to increase the perceived gradient of local changes in contrast, and this although minimized here by the use of sinusoidal gratings would also be expected to increase bout rate [14].

In response we explored the possibility that an additional mechanism of motor inhibition participates in bout initiation. The basic idea is that the probability of a bout start event is reduced in relation to the recent amount of effort expended in swimming; in effect, all else being equal, stronger bouts are followed by longer recovery periods. The weaker bouts seen at low level in the baseline flow procedure result in less inhibition, counteracting the lower

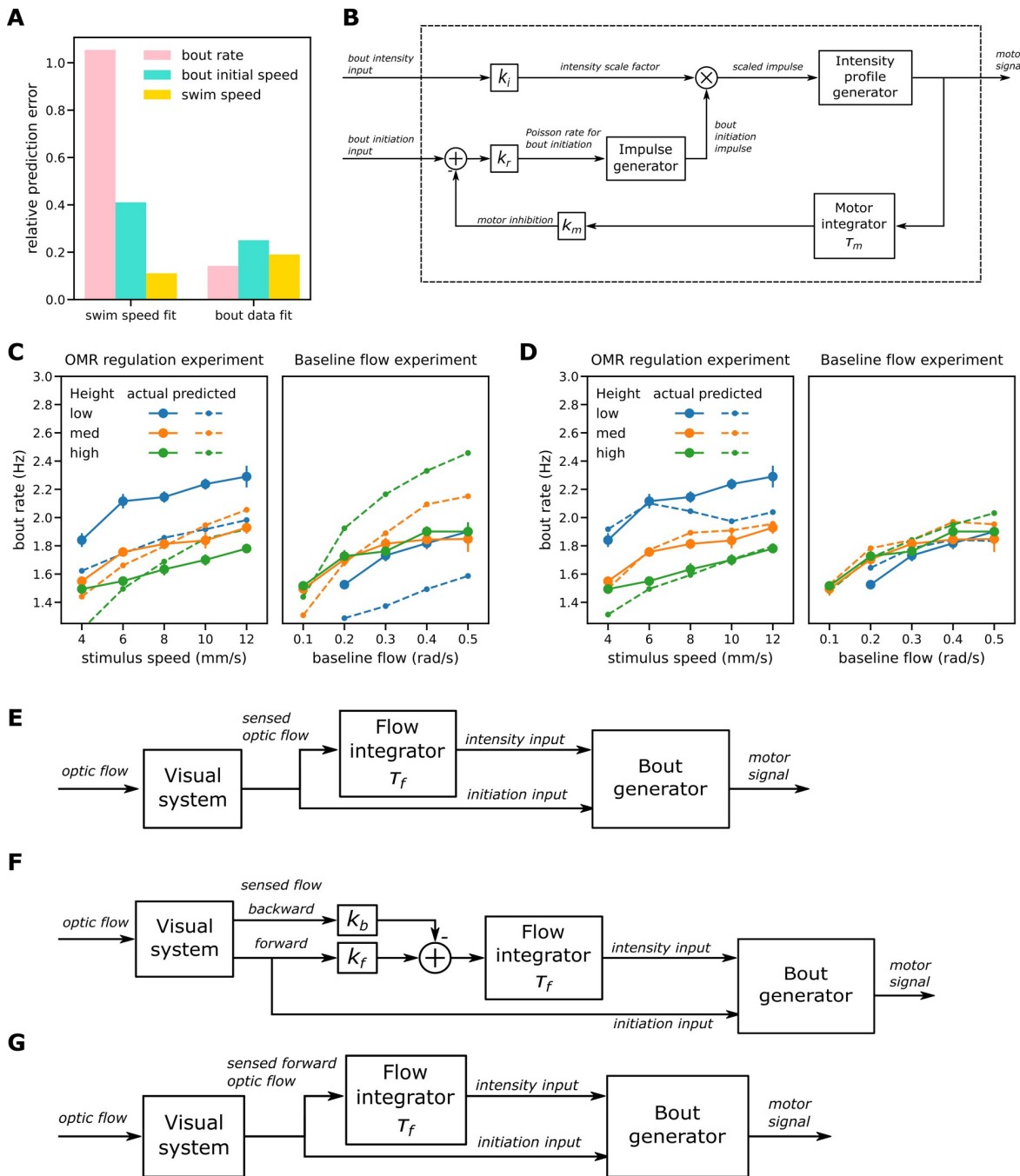

**Fig 5. Modeling II: enhanced bout generation and two-factor model.** (A) Mean prediction errors for the single factor model fitted to mean swim speed (left group of bars) and to bout measures (mean bout rate and intensity) (right group). Fitting to swim speed failed to predict the bout measures; fitting to bout data reduced prediction errors for bout measures but produced less accurate swim speed predictions. (B) Enhanced bout generator has separate inputs for bout intensity and bout initiation and the latter is subject to motor inhibition. (C) Bout rates for the single factor model fitted to bout measures. The model failed to predict that bout rates are sensitive to height in the OMR regulation procedure (left) but not the baseline flow one (right). (D) Bout rates predicted by the enhanced bout generator when fitted to bout data and constrained to reproduce the observed bout intensities. It was now possible to reproduce the bout rate patterns observed in the two experiments. (E-G) Dual factor model variants. Each variant has an enhanced bout generator with separate bout intensity and initiation inputs. Optic flow is integrated only for the intensity input. (E) Variant A: the overall optic flow is integrated. (F) Variant B: input to the flow integrator is a linear combination of the forward and backward components of optic flow. (G) Variant C: only the forward component of the optic flow participates in control of the translational OMR.

tendency to start swimming due to the greater reduction in flow sensed immediately following a swim bout and potentially resulting in similar mean bout rates at different heights. Further analysis of the experimental data gave some support for this idea: over all conditions there was a positive correlation between the strength of an OMR-consistent bout and the time from the start of that bout to the start of the next for both the OMR regulation experiment (t(83) = 10.08, p<0.001) (Fig 3G) and the baseline flow experiment (t(81) = 7.20, p<0.001) (Fig 3H)

The resulting bout generator model (Fig 5B) has separate inputs controlling bout initiation and intensity, with motor inhibition only involved in the former: the motor signal is subject to leaky integration with time constant $\tau_m$ and gain $k_m$ and the result subtracted from the bout initiation control input. In some respects this motor inhibition mechanism resembles that presented by Markov et al [12], but in their model bout initiation was based on a threshold mechanism rather than a Poisson process. Furthermore, they proposed a separate threshold for bout termination and did not model control of bout intensity: all bouts had the same fixed speed of 20 mm/s throughout and varied only in duration. In contrast, our account, because it does model swim speeds within bouts, has no need for a bout termination mechanism. Instead, following the fixed relative profile, the swim speed reaches a peak and then decays to zero without any explicit termination event.

With the addition of motor inhibition, we found parameters (Table 3) that gave a better fit to bout rate (relative error reduced from 0.105 to 0.056) and replicated the finding that height has little impact on bout rate when the baseline flow is held constant (S6B Fig). As for the basic bout generator the best fit was achieved when the flow integrator passed the instantaneous optic flow signal straight through without integration suggesting that the model can be simplified by removing flow integration altogether from the signal path for bout initiation.

**Dual factor model predicts bout characteristics as well as mean swim speed.**   Armed with the improved model for bout initiation we returned to developing a complete model that includes a process for controlling bout speed. The bout initiation input of the enhanced bout generator (Fig 5B) receives the sensed optic flow signal directly as it seems that flow integration is not required for that function. The intensity input receives an optic flow signal after leaky integration. For the initial dual factor model considered, variant A (Fig 5E), the input to the flow integrator was assumed as before to be the overall sensed optic flow. Model parameters were again found that minimize the prediction error for mean bout rate and bout speed simultaneously using the pooled data from both experiments (Table 4). Variant A performed better than the basic single-factor model when fitted to bout statistics (Fig 6A) but still failed to give a good match in some cases, in particular bout rate in the OMR regulation experiment (S6A Fig).

For variant A, the optimal time constant for flow integration turned out to be very low (0.024s) which might be expected to result in the model predicting a lower degree of regulation that is actually observed. A possible reason for finding such a low integration time constant is that otherwise the negative flow induced by swimming would result in an intensity control

**Table 3. Optimal parameters for the enhanced bout generator when predicting bout rates only.**

| Parameter | Value |
|---|---|
| $k_r$ | 1414.21 |
| $k_m$ | 0.571 |
| $\tau_m$ | 0.112 |

Initial parameter values for the enhanced bout generator found by minimizing the bout rate prediction error with the intensity scale factor fixed for each experimental condition at the value that yields the observed mean bout intensities.

signal that stays negative for a significant amount of time after the bout finishes, preventing the timely generation of bouts with non-zero intensity. At any rate, such considerations suggested a model in which backward optic flow has a less powerful influence over bout intensity than forward flow.

To explore this we generalized the two factor model to associate different gains with the forward and backward translational flow components in a similar manner to Markov et al [12]. Neurobiological evidence that different groups of neurons in the pretectum and tectum respond selectively to different directions of optic flow [35,36] suggest that such a scheme is biologically plausible. Input to the flow integrator is now a linear combination of the forward and backwards components parameterized by these different gains. As the intensity gain parameter of the bout generator is now redundant, it was fixed at 1 (Fig 5F). For simplicity the bout rate input to the bout generator is shown as receiving the forward optic flow rather than the overall flow. This makes no difference to model output as without flow integration it is only the immediate optic flow that influences bout initiation and initiation events can never occur when the flow input is negative.

With separate intensity gains for the forward and backward components of optical flow we achieved a better fit to the experimental data (Figs 6A, S7C and S7D). In fact, the best fit was achieved when the gain for backward flow was a very small proportion (0.027%) of that for forward flow (Table 4), suggesting that backward optic flow has little or no influence on bout speed as well as on bout initiation.

Accordingly, we simplified the enhanced model by setting the gain for backward optic flow to zero. For the resulting variant C (Fig 5G) the input to the flow integrator is the forward component of optic flow rather than the overall flow; it is otherwise identical to the initial variant A. Sample signal traces are shown in Fig 6D. After optimizing parameters for variant C starting from the optimal parameters found for variant B, both variants gave similar prediction errors (Fig 6A). The optimal time constant for flow integration for variants B and C was considerably greater than that for variant A, from 0.024s to 0.152s, supporting the idea that participation of backward optic flow reduces model fit for bout speed as well as bout initiation. As model C is simpler than model B but gives similar results, we selected it as the final model.

This final model suggests that some features of the OMR result from interactions between several mechanisms that can be fairly complex. To give a specific example, at first sight our empirical observation that bout rate was not affected by height in the baseline flow procedure (Fig 3D) might suggest that bout initiation depends only on baseline flow and is thus independent of behavior. Model simulation however suggests this is too simple (S9 Fig). Swim bouts in the baseline flow procedure were slower at the lower height resulting in less motor inhibition

**Table 4. Optimal parameters for the dual process model variants.**

| Parameter | Model A | Model B | Model C |
|---|---|---|---|
| $k_f$ | 1* | 291.204 | 291.204 |
| $k_b$ | 1* | 0.400 | 0* |
| $\tau_f$ | 0.0238 | 0.152 | 0.152 |
| $k_i$ | 1055.42 | 1* | 1* |
| $k_r$ | 772.934 | 274.831 | 274.831 |
| $k_m$ | 0.034 | 0.021 | 0.021 |
| $\tau_m$ | 0.345 | 0.792 | 0.792 |

Parameters fixed in advance and constituting part of the model definition are indicated with an asterisk (*). Values of the other parameters were found by simultaneous minimization of the relative bout rate and bout speed prediction errors for the experimental data (see Methods section for details)

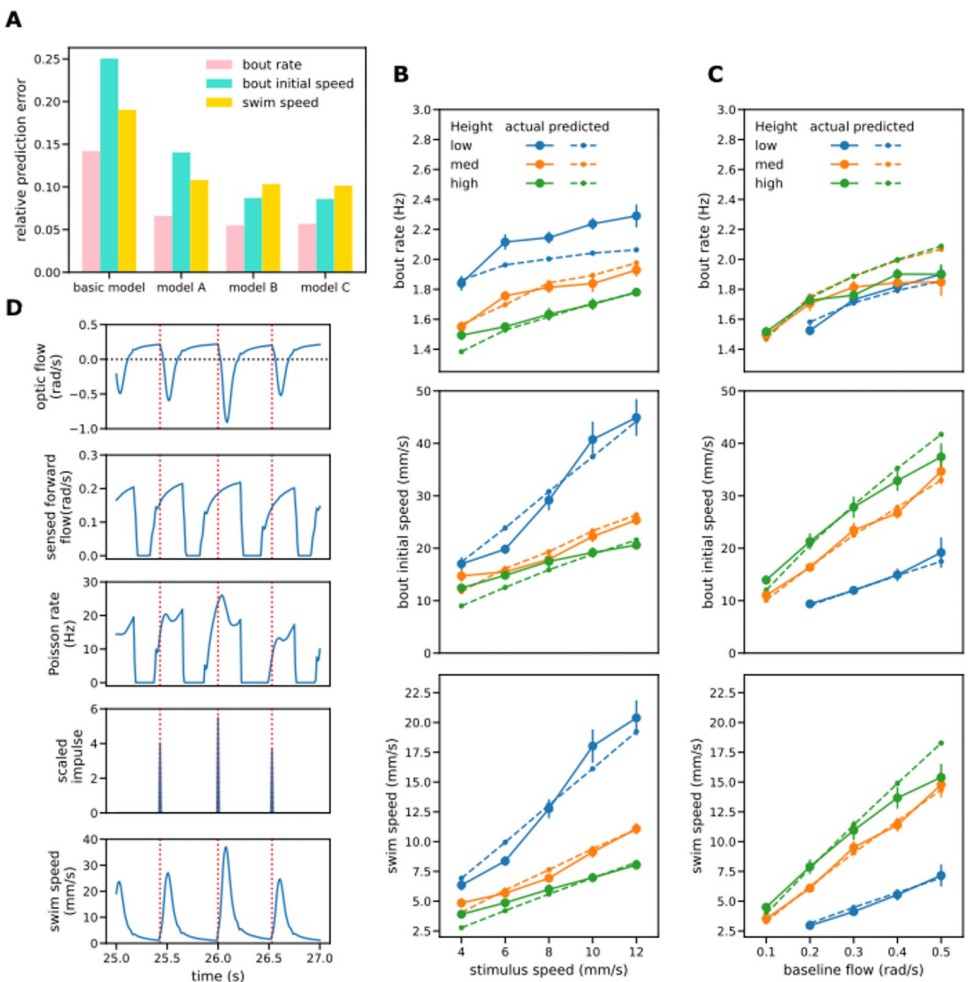

**Fig 6. Modeling III: performance of dual factor model.** (A) Relative prediction errors for the basic single factor model and the three variants of the dual factor model. All variants of the dual factor model predicted both bout measures and swim speeds better than the single factor model. Variant C that processes only forward flow gave lower prediction errors than variant A and similar ones to the more general variant B, suggesting its adoption as the preferred model. (B-C) Comparison of observed bout rates, bout speeds and mean swim speeds with those predicted by variant C of the dual process model. In general, a good fit is achieved to the observed values for both OMR regulation (B) and the baseline flow procedures (C). (D) Sample signal traces generated by variant C of the two-factor model when height is intermediate (32mm) and stimulus speed 8 mm/s. Red dotted lines indicate times of bout initiation.

and a tendency for the next bout to start sooner. On the other hand, at the lower height the bouts were still not sufficiently slow to compensate completely for the much lower stimulus speed. Consequently, swimming during the bout depressed optic flow below its baseline to a greater extent which tended to increase the interval to the next bout. The net result of these opposing effects was a bout rate that is similar at both heights.

To sum up, comparison of observed bout rates, bout speeds, mean swim speeds and degrees of regulation with those predicted by the final model showed a generally good fit for all measures and for both for the OMR regulation procedure (Fig 6B) and baseline flow procedure (Fig 6C). In particular the final model reproduced two key qualitative experimental findings: first, for a given stimulus speed, all four measures including degree of regulation vary systematically with height; second, for a given baseline optic flow, bout rate is relatively independent of height.

## Discussion

### Conclusions

The final model (Fig 5B and 5G) suggests that the translational OMR involves two control processes, the first influencing the start time of an individual swim bout and the second determining the overall speed or intensity of that bout.

The bout initiation process is driven by the immediate sensed forward optic flow without integration in conjunction with an effect of motor inhibition that tends to delay the start of a bout to an extent depending on the strength of the previous one. The sum of these two influences determines the probability that a bout will start at any given time. As the fish typically swims much faster than the stimulus for the active part of the bout, the optic flow is initially negative and only becomes positive again after swim speed falls to below that of the stimulus. After the fish detects this return to forward flow, the probability of initiation of the next bout (its Poisson rate) starts to rise steadily from zero towards an asymptotic value determined by the baseline optic flow as the fish's speed through the water and the motor inhibitory effect of performing that bout both decline (Figs 6D, S8 and S9).

The process for bout speed control also depends on the sensed forward optic flow, but now leaky integration is involved: the level of the integrated flow at the time a bout is initiated determines the initial speed of that bout. Because instantaneous swim speeds have a fixed relative profile whose form is independent of the overall bout speed, the initial speed is then sufficient to define the instantaneous swim speed at all times from the start of this bout to the start of the next.

The good fit of this model to data (Fig 6A–6C) shows that these two processes acting together can account for the relationships observed here between stimulus speed and height, on the one hand, and mean bout rate, bout speed, overall mean speed and degree of regulation on the other. The model therefore seems well suited to describe the control algorithms underlying the effects of optic flow on the translational OMR.

Experiments and modelling together suggest a number of new conclusions about the translational OMR:

**The degree of OMR regulation is only partial and varies systematically with height.** In experiment 1, which simulates the natural situation in which a fish may swim at different levels against an approximately constant water stream, some degree of regulation was observed. However, it was far from perfect, with mean swim speeds varying systematically with height: during OMR trajectories fish tended to swim faster than the stimulus ("over-compensate") when close to the bottom and slower ("under-compensate") at high levels (Fig 2E).

**Swim bouts have a relative bout speed profile that is independent of their overall speed.** (Figs 3F and S3). It is also noteworthy that the profiles for OMR-consistent bouts were essentially identical to those observed when the direction of swimming was not taken into account, suggesting that the bout generation mechanism proposed here may also underlie behaviors other than the OMR (Fig 3E).

**Bout initiation and bout speed are separately controlled.** In experiment 2, the free-swimming analogue of VR procedures that vary feedback gain while holding the baseline stimulus speed constant, we found that bout speed reduces as height reduces (gain increases) but mean bout rate remains unchanged (Fig 3D). In experiment 1, bout speed and rate changed together (Fig 2F). This dissociation suggests that different mechanisms must underlie the control of bout initiation and bout speed. Modeling reinforces that conclusion and suggests specific mechanisms underlying those two control processes.

**Variations in bout speed and active duration may share a common mechanism.** The fixed relative speed profile predicts correlations between bout speed and bout duration (i.e.,

the duration of active swimming preceding the interbout interval) as reported in previous studies [25] and observed here (S4 Fig) with the details depending on how bout termination events are identified (e.g. swim speed or tail curvature falling below some threshold). This suggests that bout speed and active duration may not be independently controlled. Modelling again reinforced this conclusion: control of bout initiation and initial speed alone was sufficient to predict mean swim speed, suggesting that any independent process controlling bout duration could have limited impact at least for the slow swim bouts observed here (Fig 6B and 6C)

**Only forward optic flow plays a significant role.**   The results of modeling suggest that the effective external stimulus for the translational OMR is restricted to the forward component of the whole-field translational optic flow, with backward flow playing no significant role in the control of either bout initiation or speed. This may seem surprising given the common assumption that the OMR serves to minimize drift, since the backward optic flow provides information that should make good regulation easier to achieve, but is consistent with the finding that regulation is often quite poor.

## Characterizing the OMR

The optomotor response is sometimes called the "optomotor reflex" [5,37], though more often when referring to the rotational rather than the translational component, or characterized as a "position-stabilizing reflex" [11]. However, the results presented here suggest that at least the translational component of the OMR is not a typical reflex. In a reflex action, "neural activity evoked by stimulation of sensory receptors is transmitted directly to response production mechanisms" [38]. However, the translational component of the OMR, a behavior that continues over a sustained period of time, is not a direct response to an immediate eliciting stimulus but requires intermediate processes that involve integration and therefore a form of memory both of the optic flow for control of bout intensity and of motor activity for control of bout initiation.

It has also been suggested that the translational OMR is an adaptive locomotor behavior that involves a form of motor learning [9,10] but the current model suggests that its control, at least over relatively short time scales, does not require learning–at least if learning is defined as a persistent change in behavior brought about by exposure to specific environmental events during the organism's lifetime. Similarly, as Markov et al argue [12], "adaptation", which implies a persistent change in the brain in response to environmental or bodily changes, does not seem to be required for the moment-to-moment control of the OMR.

Instead, the OMR is probably an innate sensorimotor behavior that emerges through development. Swimming behavior during individual bouts in some ways resemble a traditional "fixed action pattern" [39] or "modal action pattern" [38,40] whose release and overall vigor depends on external stimulation while the detailed pattern of behavior takes a stereotyped form typical of the species that is generated intrinsically rather than determined by such stimulation. The OMR as a whole might then be characterized in ethological terms as a stream of modal action patterns of varying intensity although in this case the immediate releasing stimuli are internal rather than external and generated through processing of the whole-field forward optic flow and a motor-related signal.

## Neural mechanisms underlying the OMR

As an algorithmic level model [41] the current model has nothing directly to say about how component operations are implemented at the circuit level or map to different brain regions,

but it does both constrain and suggest possibilities for an underlying neural implementation. For the latter, one possibility is that the visual processing, including integration, is a function of pretectal regions, generation of bout initiating impulses a function of midbrain or hindbrain areas, and that central pattern generators (CPGs) located in the rostrocaudal spinal cord are responsible for the detailed pattern of tail movements. There is neurobiological evidence for the first [11,12,14,35,36] and last of these [16,42] but less evidence relevant to the proposed intermediate processes. Outstanding questions include how the magnitude of the logical impulse received by such a CPG is encoded at the neural level. For example, each bout might be triggered by a brief burst of firing where some combination of duration, frequency, and/or the number of neurons involved corresponds to the magnitude of the scaled impulse in the algorithmic model. Another is the locus of motor integration. One possibility, as suggested by the block diagram for the bout generator, is that it is collocated with impulse generation, but it could equally be located in motor or even sensory areas.

## Function of the OMR

The prevailing account of the function of the OMR is that it serves to hold position relative to the ground [13]. Here we find that a degree of regulation certainly occurs in the analogous experimental situation, but it is only partial and depends upon height: for a fixed stimulus speed, mean swim speeds decrease markedly as height increases. This suggests that the prevailing account may be an over-simplification at best, but there are many other factors to be considered especially in the natural setting.

First, there is the possibility that in 6–7 dpf larval zebrafish the OMR is still developing with regulation improving later in life, although the observation that older zebrafish larvae (16–18 dpf) and adults tend to exhibit a negative translational OMR [13]–i.e. swim along with the current rather than against it–makes that seem less likely. It is also possible that OMR behavior varies widely between individuals, with only a few achieving good regulation. The vast majority of individual larvae do not reach adulthood for many reasons including being swept downstream; perhaps those that survive are mostly good regulators for whom the OMR fully achieved its assumed function of holding position.

Second, note that we do not see either over- or under-compensation as necessarily problems for the fish: here we use these terms merely to indicate deviations from the ideal performance of a regulator that tries to match swim speed to stimulus speed. In both natural and experimental settings the OMR may occur in alternation with other behaviors with an impact on overall regulation, as observed for prey capture in some insects [43]. A regulator that achieves perfect speed matching during periods when it is operating thus may not be optimal overall: some over-compensation during the OMR could improve overall position holding when other behaviors cause the fish to fall back with the stream, and the converse is true for behaviors that involve fast swimming into the stream such as prey capture.

Third, in order to focus on the control processes underlying the relationship between optic flow and the OMR, in our experiments we minimized the presence of other visual cues. In natural settings, such cues may be available, such as local edges [14] or textural features of the ground image whose salience changes with height, and affect the relationship between height and degree of OMR regulation. In addition, hydromechanical cues absent when testing in still water with a moving stimulus are present in moving water with a stationary stream bed and may play a role. Such cues can be detected by the lateral line system and in the absence of vision are sufficient to elicit a degree of counterflow swimming, although once counterflow swimming has begun, for larval zebrafish at least, the degree of regulation in the presence of

visual cues seems to be unaffected by the presence of lateral line stimulation [44]. In a natural setting the degree of regulation achieved by the OMR may thus be less sensitive to the impact of height on optic flow than in the experimental one.

Finally, the OMR in zebrafish larvae may serve functions in addition to or even instead of reducing drift. One possibility is to coordinate the swimming of neighboring individuals: all will tend to swim in the same direction when subject to a similar water current. (There is some evidence for a similar effect of the rotational OMR in the adult medaka [45], though not in larvae). This may tend to keep individuals together reducing vulnerability to predation at a time when shoaling behavior has not yet developed. Other suggestions include energetic benefits of swimming into the stream and enhanced escape from suction predators [13,44]. Finally, swimming is required for individuals to come into proximity with food and the OMR promotes a swimming direction that maximizes the speed of approach to upstream targets while simultaneously reducing drift. In general, position holding is unlikely to be the only goal. For example, when feeding close to the stream bed, more frequent proximity to food particles may be more important than position holding and an over-compensating OMR beneficial overall even when it results in swimming somewhat faster than the stream.

Fieldwork and further experimentation would be required to clarify the extent to which the OMR in fact holds position in natural settings, its sensitivity to height when cues other than optic flow are available, and the way that the OMR interacts with other activities such as prey capture, predator surveillance and exploration which may themselves be affected by height above the stream bed.

## Other model variants

The current model proposes a non-homogeneous Poisson process for generation of initiating impulses and that motor inhibition results from integration of a motor signal that reflects the observed swim speed. Other variants are of course possible. For example, with the involvement of motor inhibition a mechanism that generates a spike whenever the sensed optic flow first exceeds the inhibitory motor signal by some threshold might be sufficient. The inhibiting signal might result from leaky integration of the scaled impulse rather than a more downstream motor signal, giving a faster response that might even remove the need for a separate refractory period. Future work includes exploration of such options.

Modeling the input signal to the intensity profile generator as an impulse is the simplest possible approach as it allows the impulse response of the profile generator to be recovered directly from observed behavior. In reality, that is likely to be an oversimplification: as suggested above when considering the neural implementation, bout intensity may reflect the duration of an initiating pulse instead of or as well as its magnitude. Recovering the impulse response from observed behavior would then require deconvolution and knowledge or assumptions about the form of the input signal, including incorporation of relevant local features [14] as well as whole-field optic flow.

## Are swim bouts always ballistic?

Some other studies have been interpreted as suggesting that swim bouts are not always ballistic but may be modified by external stimulation once past the initial period of sensory delay. For example Markov et al [12] observed faster swimming after the initial 220ms of a bout in a VR situation for open loop bouts, when grid movement was unaffected by swimming, than for closed loop bouts. They suggested that bouts consist of an initial ballistic period followed by a subsequent reactive period. In contrast, the current model suggests that the original bout is entirely ballistic; instead, continued exposure to forward optic flow in the open loop condition

triggers a second bout whose impact on swim speed merges with that of the first. Further research is required to decide which interpretation is correct.

## Methods

### Ethics statement

All procedures were in accordance with the UK Animal Act 1986 and were approved by the Home Office and the University of Sussex Ethical Review Committee under the project license PPL1039528.

### Animals

AB wild-type zebrafish (Danio rerio) adults were housed in the aquatic facility in the University of Sussex. All experiments were performed on larvae aged 6 to 7 days post-fertilization (dpf). Larvae were reared in Petri dishes in E2 solution on a 14/10 hour light/dark cycle at 28˚C. Pebbles were added in Petri dishes and growing larvae were put above white noise grids for habitat enrichment and early visual feedback.

### Free-swimming behavioral assay & procedures

We used a free-swimming assay to mimic the natural environment in which larval zebrafish swim against aquatic perturbations. Larvae were put in 300mm long clear acrylic channels with width 10mm and depth 8mm. Eight larvae were tracked simultaneously in parallel channels with visual barriers so that the fish could not see each other. Moving sinusoidal gratings with 100% contrast were presented from below to simulate the optic flow experienced in streams. Height was varied by adjusting the stimulus monitor within the range of 4-60mm. Position of the fish was tracked through processing the 2D image taken from above at 100 frames/s using Basler acA1920-155um USB 3.0 camera with a resolution of 2.3 MP. The setup was illuminated by IR light at a 45 degree angle from above on the side.

Grating speeds were generally kept relatively low to avoid eliciting escape behaviors. The period of the grid was varied in proportion to height for the three height conditions to maintain the same spatial frequency of 0.0156 cycles per degree (cpd) at the center of the channel when refractive effects are ignored, as the frequency eliciting fastest swimming at the intermediate height was found to lie between 0.01 and 0.02 with relatively little variation across that range (S10A Fig). In fact refraction effects increase the perceived spatial frequency to an extent that can vary with height [27]. However, a further experiment demonstrated that the same grid period scaled in proportion to height was also optimal at both the low and high heights (S10B Fig), suggesting that in these experiments the impact of variations in perceived spatial frequency due to changes in height had a limited effect on the OMR.

Calculating the angle subtended at the eye by rays from a single period of the grid stimulus located directly below the fish indicated that the perceived spatial frequency would have varied from 0.0187 cpd for a fish in the low height condition to 0.0210 cpd in the high height group (S11 Fig). The spatial frequency tuning curve (S10A Fig) suggests that this degree of uncontrolled variation (± 7.7% relative to midpoint of the range) around a point close to the peak of the curve would not have been expected to have much effect on OMR performance, as confirmed by the subsequent experiment. In addition, because refractive effects are at a minimum immediately below the fish but increase markedly toward the periphery, there is some suggestion that the fish may have been responding primarily to optic flow from the region of the ground immediately below, as they do when responding to local edges [14].

Each experimental session consisted of three traverses at the same height. The initial traverse was to drive the larvae to one end of the channel with the stimulus moving at 5 mm/s for 70s. During the two *trial traverses* the grating moved first forward for 60s then backward for 60s at the appropriate speed for that session with a resting period of 5s between trials. Experiments were conducted on two consecutive days with larvae aged 6 and 7 dpf respectively. The monitor presenting the stimulus was manually adjusted for the three experimental heights at 4–12, 28–36 and 52–60 mm respectively. We took a mixed design approach: each batch of 32 larvae (total of 96 per experiment) was subjected to only one specific height but all the stimulus speeds. The order of experimental conditions with different combinations of testing height and speed was randomized on the day.

## Experimental data processing

**Tracking algorithm.** Algorithms were developed in C++ to determine the position of the center of mass of the swim bladder for each fish. A background image was initialized as the average of the first 2000 frames and subsequently updated at regular intervals during each session. For each frame, the image of each swim channel was processed in parallel in real time during image acquisition. The corresponding portion of the current background image was first subtracted from each channel image. The result was rectified, smoothed with a median 3x3 filter, eroded using a kernel size 2, pixel values scaled up to maximize contrast, and then thresholded to yield a binary image. Contours were extracted from the binary image with the border following algorithm of Suzuki and Abe [46] as implemented in OpenCV [47] and their centers of mass calculated and recorded for subsequent off-line analysis. As multiple contours were sometimes found in the same frame, a custom Python script using Markov chains was applied in a post-processing step to determine in this case which of the resulting candidate positions to accept as the most likely true position given the previous history.

**OMR trajectory extraction and mean swim speed.** Position data from the tracking algorithm were first resampled with a fixed timestep of 10ms. Time gaps of up to 120 ms were filled by linear interpolation and the rest left as missing data.

The focus here is modelling the algorithms underlying the OMR, and as far as possible we wanted to exclude data from periods with a mixture of different behaviors. The criteria for including a period for analysis were therefore chosen to be quite strict: an *OMR trajectory* was defined as a time series of duration at least 1s with no remaining missing values during which the mean swim speed was at least 0.5 mm/s and the direction of swimming was the same as that of the stimulus throughout. Times at which swimming changed direction were defined as zero crossing times for the difference between two exponential moving averages (EMAs) of position. The EMA at time t for positions $x_0 \cdots x_t$ with smoothing factor α was calculated using the *ewm()* method in pandas [48] as

$$y_t = \frac{x_t + (1-\alpha)x_{t-1} + \cdots + (1-\alpha)^t x_0}{1 + (1-\alpha) + \cdots + (1-\alpha)^t}$$

α was 0.002 for the slow decay EMA and 0.05 for the fast decay EMA. The direction of time was reversed when looking for the end point of a trajectory to ensure that swimming was in the appropriate direction throughout.

The resulting OMR trajectories were pruned to exclude timesteps following the first arrival at a point 50mm before the physical end of the channel for each of the two trial traverses in an experimental session.

The mean swim speed for each larva during a trial traverse was calculated as the total distance moved during the OMR trajectories extracted from that traverse divided by the sum of

durations of those OMR trajectories. Data were discarded from traverses whose total OMR trajectory duration was less than 2s. The overall mean swim speed for an experimental session was calculated as the mean of the mean speeds for its two experimental traverses; if neither traverse had a total OMR trajectory duration of at least 2s then the data from that session was discarded.

**Bout extraction.** A low-pass bidirectional filter with a 'kaiser' window which has a length of 20 and cut-off frequency of 0.02 was applied to the position to smooth the data. Derivative functions were then used to obtain the velocity and acceleration. The bout onsets were defined as local maxima of acceleration. Only bouts with a duration of more than 50ms and a distance of at least 0.5mm were considered valid.

**Relative bout speed profile.** Instantaneous swim speeds were calculated as the gradient of the position data by taking first order differences without applying smoothing. These swim speeds at time offsets up to 1s (100 timesteps) from bout start were averaged over all bouts that occurred during the two trial traverses to give the mean absolute speed at each time offset. These absolute mean speeds were then normalized by dividing by their mean to yield the overall relative speed profile. Other profiles were calculated in the same way after selecting timesteps appropriately–for example, only timesteps belonging to OMR-consistent bouts when calculating the relative speed profile for OMR bouts.

## Statistical analysis

R [49] was used for statistical analysis. Welch F tests were used to assess the impact of height on the dependent variables (swim speed, OMR ratio, bout rate and bout intensity) as the requirement of homogenous variance for classical Fisher ANOVA was not met, with observations for the repeated measures variable (stimulus speed in experiment 1, baseline flow in experiment 2) averaged for each individual.

## Model description

**Optic flow and sensory model.** Movement of the fish relative to stimulus below generates an optic flow ω in radians/s:

$$\omega(t) = \frac{1}{h}(s - v(t))$$

where $v(t)$ is the speed of the fish through the water at time $t$, $s$ the forward speed of the grid stimulus (this corresponds in the natural setting to the speed of the water current sweeping the fish backwards), and $h$ the height of the fish above the water (Fig 1A). Note that height and stimulus speed, although varying between experimental conditions, are fixed for a given trial while the swim speed and resulting optic flow vary. The convention here is that optic flow is positive when the fish is at rest with the stimulus moving and is reduced by swimming. The output of the fish visual system, the sensed optic flow $y(t)$, is assumed to be an accurate estimate of the actual optic flow but with a sensory delay $t_{delay}$ of 220ms:

$$y(t) = \omega(t - t_{delay}) = \frac{1}{h}(s - v(t - t_{delay}))$$

**Leaky integration.** Integration is assumed to be subject to a "leak" that causes output to decay exponentially to zero in the absence of input. Optic flow integration is modeled in

continuous time by the ODE

$$\frac{dY(t)}{dt} = y(t) - Y(t)/\tau_f$$

where $Y$ is the output of the integrator and $\tau_f$ is the decay time constant. This is equivalent to a low pass IIR filter with time constant and gain both $\tau_f$; for a fixed input $y$ the output $Y$ relaxes exponentially toward asymptotic value $y\tau_f$.

For model simulation the discrete time approximation uses Euler integration with timestep number $t$ and time step duration $\Delta t$ of 10ms:

$$Y[t] = Y[t-1] + \Delta t \left( y[t] - \frac{Y[t-1]}{\tau_f} \right)$$

**Single factor model.**   For the single factor model, the output of the flow integrator is the only input of the bout generator and governs both bout initiation and bout intensity. Within the bout generator, bout initiation events are represented by a train of unit impulses. These are emitted by an impulse generator implementing a nonhomogeneous Poisson process whose instantaneous rate is $max(0, k_r Y(t))$ with $k_r$ a fixed model parameter and output $z(t)$. To avoid immediate re-triggering we also require a minimum period of 250ms, the refractory period, between consecutive impulses.

The intensity of a bout initiated at time $t_0$ is determined by $k_i Y(t_0)$ with $k_i$ another fixed model parameter. The intensity profile generator is modeled as a linear time invariant (LTI) system. Its input is the train of scaled impulses $k_i Y(t)s(t)$ and its impulse response $g(t)$ can be estimated directly from the experimental data input. Its output is

$$m(t) = \int_0^t k_i Y(\tau)z(\tau)g(t-\tau)d\tau$$

For the corresponding discrete time approximation for simulation of the impulse generator, the probability that a swim bout is initiated on timestep $t$ is

$$p[t] = min(max(k_r \Delta t Y[t]), 0), 1)$$

The profile generator LTI for simulation has impulse response $g[t]$ and we simplify by assuming that speeds during a bout are independent of the history of previous bouts and thus determined only by the magnitude of its initiating impulse. The motor output is therefore simply

$$m[t] = max(k_i Y[t_0]g[t-t_0]), 0)$$

where $t_0$ is the time step at which the most recent bout began.

The motor system translates the motor signal to swimming movements that generate a propulsive force resulting in movement through the water. Adopting the simplest possible physical model, we assume that inertia is negligible relative to viscous friction and that there is no delay associated with the motor system so that swim speed through the water $w$ is proportional to the instantaneous propulsive force and that propulsive force is proportional to the motor signal. In addition, the model does not attempt to account for the effects on performance of variables such as fatigue, water temperature and water viscosity, factors that are held as constant as possible during the experiments. We therefore assume both proportionality coefficients are also constant and thus can be absorbed into the existing parameter $k_i$ to identify

swim speed $f$ directly with motor output:

$$f[t] = max(k_i Y[t_0]g[t - t_0], 0)$$

**Dual factor model with motor inhibition.**   For the general two-factor model the visual system is assumed to provide separate equally delayed accurate estimates of forward and backward optic flow, $y_f(t)$ and $y_b(t)$, with the convention that both are positive so the overall sensed flow is $y_f(t) - y_b(t)$. The input of the flow integrator is a linear combination of the two with positive coefficients $k_f$ and $k_b$ so the output of the flow integrator in the discrete-time simulation is

$$Y[t] = Y[t - 1] + \Delta t \left( k_f y_f(t) - k_b y_b(t) - \frac{Y[t - 1]}{\tau_f} \right)$$

The associated bout generator (Fig 5B) has two inputs. The first input is driven from the optic flow integrator output and determines bout intensity while the second is driven directly by the forward component of sensed optic flow and influences bout initiation. The generation of bout initiation impulses is inhibited by an additional motor feedback loop that reduces the instantaneous Poisson rate. For the corresponding discrete time approximation, the output of the motor integrator with leak time constant of $\tau_m$ is

$$M[t] = M[t - 1] + \Delta t \left( m[t] - \frac{M[t - 1]}{\tau_m} \right)$$

The probability of bout initiation during timestep $t$ is

$$p[t] = min \left( max \left( \Delta t(k_r y_f[t] - k_m M[t]), 0 \right), 1 \right)$$

where $k_m$ is a model parameter. As before, bout initiation is subject in addition to the refractory period constraint.

The swim speeds for a given intensity input follow the same equation as that for the initial model:

$$v[t] = max \left( k_i Y[t_0]g[t - t_0], 0 \right)$$

Three variants of this general model are considered (Fig 5E–5G). For variants A and C, $k_f$ is set to 1, $k_b$ is set to 1 or 0 respectively, and $k_i$ is fitted to data; for variant B $k_f$ and $k_b$ are fitted to data and $k_i$ is set to 1.

## Simulation and model fitting

**Model simulation.**   Bespoke software was developed in Python for agent-based simulation of the discrete-time model approximation described above. The timestep was 10 ms, the same as the interval between camera frames in the experiments. Each session consisted of 30 s exposure to the simulated moving stimulus, with the final 20 s of behavior analyzed, giving time for a steady state to be reached during the initial 10 s. 30 simulated fish were run for each condition using the same values for the independent variables (height and stimulus speed) as in the two experiments.

**Cost function.**   To fit the model to data requires minimizing a cost, here the *relative mean prediction error* for one or more outcome variables, with respect to the model parameters. For example, when considering the single outcome variable of mean swim speed we define the individual prediction error for a given experimental condition (combination of height and stimulus speed) and parameter set as the difference between the mean swim speed observed

for that condition and that predicted through model simulation with those parameters. The overall prediction error for an outcome variable is then defined as the root mean square of the individual errors taken over all experimental conditions and the relative prediction error as the overall prediction error divided by the mean value observed for that outcome variable over all experimental conditions. These relative prediction errors are dimensionless and thus no longer sensitive to the subjective choice of units for an outcome variable (e.g., mm/s vs m/s for speed). When fitting for multiple outcome variables simultaneously (usually bout rate and bout intensity) the mean of the relative prediction errors for each variable yields a scalar cost value as required for optimization in a way that balances in an objective way the importance given to each variable.

**Optimization process.**  The optimization process was a semi-automatic method based on repeated grid search. At each iteration an n-dimensional rectangular grid of variables was generated and each point of the grid transformed to a set of model parameters. Relative prediction errors were calculated for each parameter set and the optimal set taken as the central point for the grid for the next iteration, resulting in a series of iterations with the grid translated at each step until the best point remained in the center. The pitch of each dimension of the grid was then reduced and the process repeated, eventually closing in on an optimal set of model parameters. Decisions about how quickly to reduce the pitch of a dimension were guided by contour maps and influenced by the sensitivity of the prediction error to changes in that dimension. The maps also helped to avoid settling for a local minimum when it seemed possible that a better point might lie somewhere else within the region currently being explored.

**Data and code repository.**  Experimental and modelling data and the Python code for experimental data analysis and simulation are provided in a Github repository at *https://github.com/johnholman/omr-algorithms*.

## Supporting information

**S1 Fig. Directionality index.** Proportion of swim bouts in the same direction as the stimulus for the OMR regulation experiment (left) and baseline flow experiment (right).
(PDF)

**S2 Fig. Overall OMR ratio.** Overall OMR ratios for the OMR regulation experiment calculated over the full trial period up to when the fish first reached the end of the channel rather than during OMR trajectories only. As for OMR performance during OMR trajectories, the OMR ratio decreases systematically with height. As expected, the overall OMR ratios are lower than those observed during OMR trajectories, with both high and medium height groups swimming more slowly on average than the stimulus (OMR ratio less than 1) while the overall OMR ratio for the low height group averaged over all stimulus speeds does not differ significantly from 1.
(PDF)

**S3 Fig. Bout speed profiles.** Separate relative speed profiles for bouts split into quartiles by measures of overall bout intensity other than bout initial speed: peak bout speed (A) and total bout displacement (B). The relative speed profiles are again essentially independent of bout intensity whichever measure is used and the profile for OMR-consistent bouts is similar to that for all bouts taken together.
(PDF)

**S4 Fig. Bout duration.** Bout duration for the OMR regulation and baseline flow experiments. The fixed bout speed profile gave rise to bout durations that increased with stimulus speed and baseline flow. Here bout ending was defined as the point during a swim bout when the swim

speed, averaged over a 50 ms window, first dropped below 5 mm/s. The OMR regulation results are consistent with, and reinforce, those reported by Severi [25], who found that bout duration estimated from measures of tail curvature increased steadily with stimulus speed for this range of low to moderate speeds.
(PDF)

**S5 Fig. Baseline flow results for moderate stimulus speeds.** Bout rate and initial speed for the baseline flow experiment, showing only experimental conditions with stimulus speed at least 3 mm/s and at most 15mms to exclude most spontaneous bouts (more likely to occur at slow stimulus speeds when the OMR may not be fully engaged) and fast bouts (mostly occur for speeds faster than this). For a given baseline flow, there is no suggestion that bout rate depends on height. Unlike the OMR regulation procedure, for this procedure the impact of height on mean swim speed is due to variation in bout speed but not bout rate, suggesting that control of bout initiation and bout intensity are decoupled.
(PDF)

**S6 Fig. Bout rates predictions for basic and enhanced bout generators.** (A-B) Model predictions for bout rates when bout intensity is set to the mean observed values for each experimental condition. (A) Basic bout generator. For the baseline flow procedure, observed bout rates are insensitive to height but the basic model cannot reproduce this finding. (B) Enhanced bout generator with motor inhibition. The qualitative findings can now be reproduced.
(PDF)

**S7 Fig. Performance of two-factor model variants A and B.** (A-B) Comparison of observed bout rates, bout intensities and swim speeds for the OMR regulation procedure (A) and baseline flow procedure (B) with predictions from model variant A. (C-D) Comparison of observed bout rates, bout intensities and swim speeds for the OMR regulation procedure (C) and baseline flow procedure (D) with predictions from model variant B.
(PDF)

**S8 Fig. Signal traces from the dual process OMR model: height and the OMR regulation procedure.** Sample traces for key variables of the final model (dual process model variant C) when stimulus speed is 8 mm/s in the low height (A) and high height (B) condition, corresponding to baseline flows of 1 radian/s and 0.14 radian/s respectively. Red dotted lines indicate the time of bout initiation. Bout rate and bout speed are both greater for the lower height.
(PDF)

**S9 Fig. Signal traces from OMR model: height and the baseline flow procedure.** Sample traces for key variables of the final model (dual process model variant C) when baseline flow is 0.3 radians/s in the low height (A) and high height (B) condition, corresponding to stimulus speeds of 2.4 mm/s and 16.8mm/s respectively. Red dotted lines indicate the time of bout initiation. Bout speed is lower for the lower height, but bout rates are about the same.
(PDF)

**S10 Fig. Spatial frequency and the optomotor response.** (A) Preliminary experiment to find the spatial frequency (in cycles per degree) of the sinusoidal grid stimulus most effective in eliciting an optomotor response when testing at the medium height (28-36m) with a stimulus speed of 5 mm/s. 24 7dpf AB wildtype zebrafish larvae were tested at each spatial frequency, with the grid moving first forward and then backward for two test traverses each 60s in duration. The *optomotor index* shown here is the ratio of total distance swum in the direction of the grid stimulus divided by distance moved by the stimulus. It is similar to the OMR ratio measure used elsewhere except that it is based on swimming over the whole trial period rather

than only during OMR trajectories. The spatial frequency (0.0156 cpd) used in all subsequent experiments was close to the optimal value of 0.015 found here. The absolute grid period was varied in proportion to height to keep it as constant as possible. (B) A follow up experiment checked that the same spatial frequency is also effective at the low (4-12mm) and high (52-60mm) heights. There were 8 larvae in each experimental condition, with each individual larva experiencing a single height and all spatial frequencies. The frequency of 0.15 cpd was again at least as effective as either 0.01 cpd or 0.02 cpd, suggesting that any variation in perceived grid spatial frequency due to differences in the degree of refraction experienced at different heights did not have a material impact on the strength of the OMR in these experiments.
(PDF)

**S11 Fig. Spatial frequency and refraction.** The leftmost bar shows the "nominal" spatial frequency of the grid stimulus that would been perceived in the absence of refraction for all three height conditions. However refractive effects increase the perceived spatial frequency to an extent that depends on the height of the channel above the stimulus (righthand three bars). Following [27] here we treat the bottom of the channel as an air/water interface. The perceived spatial frequency was estimated by calculating the angle subtended at the eye by rays stemming from the limits of a single stimulus period centered directly below. For the values used in these experiments, the three mid-channel heights of 8mm, 32mm and 56mm above the stimulus and a nominal spatial frequency of 0.0156 cycles per degree (cpd), the perceived spatial frequency would have varied from 0.0187 cpd to 0.021 cpd. Given the spatial frequency tuning curve (S10A Fig), this degree of uncontrolled variation (± 7.7% relative to midpoint of the range) about a point close to the peak of the curve would not be expected to have much effect on OMR performance, as confirmed empirically by the subsequent experiment (S10B Fig). As refractive effects increase towards the periphery this may also suggest that the stimulus region directly below the fish has the strongest influence on the strength of the OMR.
(PDF)

## Acknowledgments

The larval zebrafish graphic used within Figs 1–4 was produced by Lizzy Griffiths (http://zebrafishart.blogspot.com/).

## Author Contributions

**Conceptualization:** John G. Holman, Winnie W. K. Lai, Paul Pichler, Christopher L. Buckley.

**Data curation:** John G. Holman, Winnie W. K. Lai, Daniel Saska.

**Formal analysis:** John G. Holman, Winnie W. K. Lai.

**Funding acquisition:** Leon Lagnado, Christopher L. Buckley.

**Investigation:** Winnie W. K. Lai.

**Methodology:** John G. Holman, Winnie W. K. Lai, Paul Pichler, Daniel Saska, Christopher L. Buckley.

**Project administration:** Christopher L. Buckley.

**Resources:** Leon Lagnado, Christopher L. Buckley.

**Software:** John G. Holman, Winnie W. K. Lai, Daniel Saska.

**Supervision:** Leon Lagnado, Christopher L. Buckley.

**Validation:** Winnie W. K. Lai.

**Visualization:** John G. Holman, Winnie W. K. Lai.

**Writing – original draft:** John G. Holman, Winnie W. K. Lai, Christopher L. Buckley.

**Writing – review & editing:** John G. Holman, Winnie W. K. Lai, Paul Pichler, Leon Lagnado, Christopher L. Buckley.

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
