## [Decision Letter · Decision Letter 0]

5 May 2022

Dear Dr Holman,

Thank you very much for submitting your manuscript "A behavioral and modeling study of control algorithms underlying the translational optomotor response in larval zebrafish with implications for neural circuit function" for consideration at PLOS Computational Biology.

As with all papers reviewed by the journal, your manuscript was reviewed by members of the editorial board and by several independent reviewers. In light of the reviews (below this email), we would like to invite the resubmission of a significantly-revised version that takes into account the reviewers' comments.

We cannot make any decision about publication until we have seen the revised manuscript and your response to the reviewers' comments. Your revised manuscript is also likely to be sent to reviewers for further evaluation.

Sincerely,

Joseph Ayers, PhD

Associate Editor

PLOS Computational Biology

Wolfgang Einhäuser

Deputy Editor

PLOS Computational Biology

Reviewer's Responses to Questions

**Comments to the Authors:**

Reviewer #1: This study investigates the adequacy of the optomotor response for stabilizing body positions and develops a computational model that accounts for some of the key features of this behavior. The optomotor response (OMR) is a reflex behavior triggered by whole-field visual motion and is found in all vertebrate species. In zebrafish it has recently been investigated in detail and provides an important paradigm for studying sensorimotor processing.

The major finding of the manuscript is that OMR stabilization performance is bad at high and low swim heights, i.e. the swimming velocity does not match stimulus displacement. This suggests that the function of this behavior is not really position “stabilization”, which is a wide-held assumption. Furthermore, the authors characterized the modulation in swimming speed, bout frequency and duration and built quantitative models for explaining and predicting OMR responses to optic flow stimuli varied in distance and speed. This model shows that swim speed “adaptation” can easily be explained by leaky integration of the perceived optic flow, while additional model features (separation of forward from backward optic flow signals, refractory period for inter-swim-bout intervals) are necessary to achieve a good model fit accounting for observed swim bout rates and intensities as well . Such a modeling approach is useful to decipher the control strategies of the behavior and a similar model has already been put forward very recently (Markov et al. 2021).

While the study nicely challenges the assumption that zebrafish achieve stabilization with OMR and provides an interesting -albeit not entirely new- modeling approach towards the OMR control algorithms, there are also a few major caveats that limit the usefulness of the model and possible interpretations regarding the biology. Therefore, the scientific advance the study provides in its present form is limited.

Major concerns:

1) Processing steps of the raw data need to be presented and results need to be embedded in the existing literature. The authors claim that the bout speed profile has a fixed form. However, previous literature showed that bout speed durations are variable (Severi et al. 2014). The authors need to consolidate the different findings and do their best to reveal the origin of this experimental discrepancy. My impression is that the recorded data is of lower spatiotemporal resolution, and the noisiness - together with differences in tested speed regimes between Severi et al.(2014) and this manuscript - might have caused the discrepancy

a) Show a raw video of an OMR bout. Show corresponding raw position trace and also the superimposed smoothened position trace. It would be helpful to illustrate the quantitative terms for OMR in this example as well. Now the reader need to search everywhere for the definition of each term.

b) Compare speed regimes and also report absolute speeds for experiment 2. Severi et al. 2014 showed there are two modes of OMR bouts, the fast bouts have fixed duration but varied speed while the slow bouts are the opposite. Is the analysed data biased towards modeling the fast bouts only? According to Severi et al., fast bouts are much less frequent than slow ones at a stimulus speed lower than 10mm/s (Severi et al., 2014). Furthermore, the author claimed "...but an invariant relative profile suggests that bout speed and duration share a common causal factor and are not independently controlled..." which is contradicted to the findings in Severi et al. that the speed and duration of fast bouts are independently modulated by stimulus speed. This discrepancy must be addressed before the modeling section.

c) To what extent do the different smoothing kernels and filters affect the result of the “fixed bout speed profile”? Is it possible that the dominant identified profile is simply caused by the smoothing?

2) It is unclear to what extent the recorded behavior (at low stimulus speed regimes) corresponds to OMR. The authors had a thresholding approach in which they counted any swims that were fast enough and in the correct direction as OMR. However, even in spontaneously swimming fish, they will sometimes swim quickly into the OMR direction. Figure. 2C (low swim height, blue) shows that at baseline flows of 0.1, 0.2 and 0.3, the swim speed was very low and the data point 0.1 was excluded because it apparently did not contain OMR (but data points 0.2 and 0.3 were included).

a) The authors could quantify an OMR directionality index: what fraction of swim bouts were performed into the “correct” direction in order to judge whether OMR on average occurred or was absent.

b) The authors should compare the absolute speed regimes (mm/s) between experiment 1 and 2. To what extent can the different stimulus regimes explain the observed behavioral differences? Is it possible that the “decoupling” of bout rate and bout initial speed (Fig. 2D) is simply due to very low stimulus speeds for the low height in experiment 2 (spontaneous swimming), whereas such low stimulus speeds (and thus the “decoupling”) were simply not used in experiment 1?

3) The model description is not detailed enough and scattered throughout the manuscript. Which parameters (and what number of them) were fitted and which ones were fixed? What are the resulting parameter values that were used in the model (e.g. the different tau_f)? This should be shown in a table. For example, it was not entirely clear to me, whether the ki (intensity scale factor) was chosen independently for each swimming height (which would make the result of a perfect model fit to swim speed rather boring) or whether the same ki was taken for all swimming heights (the good model fit would be interesting in this case, as it would suggest that there is a single/linear motor “adaption” rule that can explain different swim speeds for different swim heights in experiment 2).

4) The stimulus setup has a gap of air between display and water body. This leads to refraction artifacts that affect spatial frequency and perceived stimulus velocity (see Dunn and Fitzgerald, Elife 2020, parameter da/dw in Figure 1). This means that the experiment in Fig. 2 is potentially ill-conceived, because the claim of “same flow across heights” is not (completely) true. The authors need to calculate the distortion and replot the data. If the discrepancy of baseline flow (and spatial frequency) is less than 10-15%, then it is probably acceptable. If the discrepancy is larger, the results need to be reinterpreted and potentially experiments repeated.

Minor points:

5) Can fish perceive depth? The key assumption for the major point in this study: self-stabilization is not the primary function of OMR, is that fish have access to depth information and do not use it well for adjusting OMR speed. It seems likely that fish do not perceive depth and the differences seen in experiment 2 across swim heights are simply due to the swim-induced optic flow and the integrator according to the model (and possibly partially due to stimulus artifacts). It would help if the authors clearly stated their interpretation of the data regarding depth perception.

6) The behavioral characterization part has terminology problems which make it difficult to understand and evaluate some results in Figure 1 and 2, some are listed below:

a) How is the term "bout initiation" defined in the section: "Bout initiation and bout intensity are separately controlled"? From what I understood, the bout initiation seems to be an event. Is it quantified as bout rate? Then it would be much clearer to replace "bout initiation" with "bout rate"

b) How is mean swimming speed calculated, the total moving distance during the whole 30-second stimulus phases, or is it only calculated for the OMR trajectory?

c) How is relative speed defined? What's its unit? And how long is the time window?

d) In the definition of "OMR-consistent bout" in the result section, it says in the bracket "see materials and methods section for details", but I didn't find the corresponding method section.

7) It is nice to see the model perform (fit) well. But what mechanism in the model actually makes it work? It would be helpful if the authors could plot the perceived optic flow speed (over time) in the model across different heights and relate it to the leaky-integrator model (length of the tau_f time constant) to understand why and at what time points relative to bout initiation the OMR swimming overshoots or undershoots and how the swim vigor gets reduced (“motor adaptation” ) when feedback is large at low swim heights. If the authors illustrated this better, then it would become more clear, how this model does not require “motor adaptation”, but that the observed swim gain adaptation across heights is simply the result of a static leaky integrator that gets charged less at low swim heights due to the large effect of each swim bout on optic flow. Also the results could be discussed in the context of Bahl et al. Nat Neurosci 2020.

8) The bout rate is calculated as the frequency of bout during an OMR trajectory which require the fish swim faster than 0.5 mm/s on average. But it is possible that zebrafish larvae, due to their intermittent swimming pattern, solved the over-compensation problem with long inter-bout pause which is left out by this criterion?

9) Please fix the aspect problem in Figure 2G-H and what's the p-value of Pearson's R

10) Please refer to the figures in the discussion section, now it is really hard to find the corresponding results.

11) Fig. 1D: Swim heights should be indicated in the main text. Color usage in panel D and F do not correspond, please consider a use of color that is easier to follow throughout the manuscript.

12) Line numbers are missing on the manuscript pages. Including them would make it easier to provide specific comments.

Reviewer #2: This manuscript is of high interest to anyone using the optomotor response to elicit locomotion or neural activity in zebrafish, but is also more broadly applicable to other systems. The authors begin by investigating something that really needs deeper investigation- namely trying to understand mechanistically what is occurring during the OMR rather than simply using it as a tool to elicit a response, and to understand its potential function for the animal. The introduction is well written and well referenced. I think this is an innovative and significant endeavor, however I was confused at the outset about the gain control, which I attempt to explain below. I believe a revision to make this clearer to your readers would be valuable.

I will just come out and say, because it will be obvious anyway from my review, that I am not a modeler nor an engineer, but a biologist who uses the OMR. Thus, I am highly interested in this manuscript and highly motivated to understand it. I suggest revisions because there were several key items, upfront, that I was not able to understand, simply in the first 2 figures on which I will focus.

Results/Figures 1 and 2 comments:

I think Figure 1 panels A and B are quite clear, important, and nicely done. What is not clear to me leaving panel B is where does the gain measurement occur? Here we are talking about the angle, the optic flow, the height, and the distance, but it would be helpful to discuss gain in this schematic because I do not understand where this is coming from.

Panel C is very nice and important to show. What are the dotted lines? This is not explained in the legend.

It is stated in Panel D that gain decreases as height increases, but I don’t understand why that is the case. I just don’t understand what you’re referring to as the gain here. This is really a main issue for me in understanding the paper. It would be very helpful to explain this explicitly for the less mathematically-savvy reader like myself. I understand how changing the height changes the optic flow. Does changing the height not also change 2 other things: the perceived speed of the grating to the fish, and the perceived period of the grating by the fish? As I understand it, you are manipulating height but could, in fact, achieve the same effect by calculating what the speed and period should look like to the fish at that height and adjusting those parameters. This is all in open-loop, correct? So if it’s not a closed-loop feedback gain you’re changing, I’m still confused as to where the gain comes from. Critically, we know changing speed and period of the grating at a fixed height drastically impacts the response of the fish. In Figure 2 you are compensating for the speed as you change height, but not compensating for the grating period, and claim this keeps optic flow constant, but I don’t understand why as these two visual stimuli are not identical. Would this not be equivalent to putting a fish at fixed height and changing grating period and keeping speed constant? As the fish moves further from the stripes, with a constant width of the screen, they will appear thinner. As someone who has done that experiment for fun, I can tell you changing the stripe width with constant speed and height has dramatic impacts on the behavioral response. Is this what you mean by gain? I’m genuinely confused here and I’d love it for you to spell it out for me very clearly. There is no mention of “gain” in the results text for Figure 1 other than to point out that it changes in panel 1.D. In the results text for Figure 2 you state that “feedback gain and baseline optic flow are both inversely proportional to height”, and say you’re changing the gain, but don’t say much else about it.

Panels E and F. Is low, medium, and high here referring to optic flow, to gain, or to height? This is not clear or stated in the legend. What is “bout initial speed” and how is this measured? I could not find this in the legend or methods. You explain OMR ratio very clearly and well in the results, but is not in the methods or legend and possible should be.

Regarding OMR ratio- it’s not clear to me what the problem is for the fish to overshoot this ratio. Why should the goal be to be exactly at 1? It is obvious to see why undershooting it could be problematic, resulting in a failure to compensate for being washed downstream, but if water is being sucked quickly downstream, what is the ethological harm of staying ahead of it? Frankly I would expect that the fish can keep up or overshoot fairly well, especially at these very slow speeds in this experiment, and I don’t see this as a failure. Also, in Figure 1, given that you’re changing the height, the absolute stimulus speed is not the perceived speed by the fish here. Some more clarity on why ratios >1 are problematic from your point of view would be helpful.

Methods comments:

1. Which WT line was utilized? Differing WT lines can respond differently to OMR.

2. Is the acrylic arena clear? Can the fish see each other? Could this contribute to their visual responses?

3. What is the actual trial duration for the trial traverses? Is it also 30 seconds like the “initial traverse”?

4. The width of the channels is not specified.

5. Most importantly, the point about grating speed being kept low. This could impact your conclusions as slower and faster swimming kinematics and bout dynamics vary significantly. While this absolutely should be in the methods, it’s important to point out that this is really a sub-section of the possible OMR responses being explored, rather than the full range of speeds, and thus, the full range of bout dynamics.

Reviewer #3: Please see Attachment

**Have the authors made all data and (if applicable) computational code underlying the findings in their manuscript fully available?**

Reviewer #1: Yes

Reviewer #2: None

Reviewer #3: Yes

PLOS authors have the option to publish the peer review history of their article (what does this mean?). If published, this will include your full peer review and any attached files.

Reviewer #1: No

Reviewer #2: No

Reviewer #3: No
---

## [Decision Letter · Decision Letter 1]

26 Nov 2022

Dear Dr Holman,

Thank you very much for submitting your manuscript "A behavioral and modeling study of control algorithms underlying the translational optomotor response in larval zebrafish with implications for neural circuit function" for consideration at PLOS Computational Biology. As with all papers reviewed by the journal, your manuscript was reviewed by members of the editorial board and by several independent reviewers. The reviewers appreciated the attention to an important topic. Based on the reviews, we are likely to accept this manuscript for publication, providing that you modify the manuscript according to the review recommendations.

Sincerely,

Joseph Ayers, PhD

Academic Editor

PLOS Computational Biology

Wolfgang Einhäuser

Section Editor

PLOS Computational Biology

Reviewer's Responses to Questions

**Comments to the Authors:**

Reviewer #1: The authors generally did a good job addressing my concerns. The modeling is clearer now. The description of variables has improved and Figure 4B is very helpful. A few points remain, as listed below.

Major:

Old point 4 on refraction artifacts: I agree with the authors that the results of their additional experiments (Fig. S9) – same spatial frequency peak for all swimming heights – suggests that the issue of stimulus artifacts did not affect the validity of their modeling results. However, to make the experiment reproducible and informative for biologists interested in the psychophysics of this OMR behavior, it would be helpful to be more careful here. The authors argue that the advancing dark edge just below the fish (Kist et al.) is the strongest driver of OMR, which is true. But the existence of a spatial frequency tuning curve shows that the spatial period matters. To address my critique, the authors could calculate how much a single stimulus period (and thereby stimulus speed) at the nadir (just below the fish) gets compressed by refraction artifacts. Using this information, the authors could estimate the relative change in angular speed, which would be a lower but useful estimate of the speed changes (more to the periphery the changes will be larger, but these stimulus positions are less relevant to the fish as the authors point out). This could be done for each different swimming height (i.e. computer-programmed angular speed vs. apparent speed given the artifacts around the nadir). If this change is small, this isn’t much of a problem and could simply be stated in the methods section. If it is larger, the authors should at least mention in the methods section that an offset of the spatial frequency tuning peak (and measured OMR ratios) is likely. The true peak is likely at a position larger than 0.015 cpd. This also means that the OMR ratios in Fig. 2b could be different.

New point: Please mention the Github repository containing experimental data, modelling data and program scripts in the paper. The repository will be a great resource for anyone continuing this type of work.

Minor:

Original point 8 on “OMR trajectory” in the context of the over-compensation problem. This comment has been unclear. Let me explain in more detail what I meant: The authors only analyse their so-called “OMR trajectory, which is a ”time series...during which the mean swim speed was at least 0.5 mm/s and the direction of swimming the same as that of the stimulus". So apparently, for the entire of duration of the experiment with motion stimulus, there were periods of zebrafish inactivity which were discarded! Have the authors tried to perform an analysis in which the complete time period during with the moving stimulus was shown is included to calculate the OMR ratio? If the fraction / total amount of time of discarded data is high, then it is possible that the over-compensation problem (for low heights) identified by the authors does on average, over longer time scales, not exist for the fish? Also see line 309: "Bout rate was defined as the frequency of bouts occurring during OMR swim trajectories" So bout rate was only measured during pre-defined "OMR trajectories", not the complete time period.

In the following, line numbers correspond to those in the manuscript version with tracked changes.

l.225: value of the used spatial frequency should be indicated in the text.

l.228: refraction had little DIFFERENTIAL effect in the experimental settings, but the tuning peak might be offset altogether (cf. major point 4).

Fig. 2D, legend: Mention what your definition of OMR ratio is.

l.361 cross-reference Fig S1?

l.510: later -> earlier?

Fig. 6D: add some vertical lines across the 5 sub-panels to ease comparison of events across sub-panels?

Reviewer #3: Review of ZF-OMR Manuscript

Hohman et al. PLoS- Computational Biology -- Resubmission

The authors have made considerable effort to address all points raised in my initial review. I appreciate the extent they went to so as to better convey the nature of the modeling and the height/OMR question. This is evident in new figures (e.g. Figure 4), a more consolidated presentation of the modeling and better discussion of their findings in the context of the natural OMR behavior.

There was some good discussion within their reply concerning the challenges of finding terminology to best describe swim bouts and the vigor of swimming. I do appreciate these challenges but in regards to their now avoiding the term “intensity” in favor of “bout speed” I am confused because the bar graph in the current Figure 5 (if I am looking at the correct figure) has bars for “intensity”, “bout rate” and “swim speed” which adds to my uncertainty. They did incorporate important methodological details on the swim channel and behaviors seen therein. While video of elemental aspects of the behavior would have been nice to include (e.g. within a local portion of the swim channel), we appreciate that kinematics of the OMR behavior have previously been reported.

I remain a bit puzzled about the authors taking an “either/or” approach in their view of the OMR and a goal of disproving the “holding station” hypothesis (lines 816 – 818). I view their nice modeling and data as endorsing a holding-station component, including initial negative movement in relation to “position” when stripes begin moving, and subsequent forward progress which contributes to navigational control of location and posture: two essential behaviors to which the basic OMR response contributes. In their reply to my reviewer comments there was much good discussion about this topic and different ecological benefits of body and position control. Some of this has been incorporated into their expanded discussion of OMR and related ecological needs and this does make for a stronger work. Overall, their Discussion is fine as is: the authors are entitled to their perspective!

**Have the authors made all data and (if applicable) computational code underlying the findings in their manuscript fully available?**

Reviewer #1: Yes

Reviewer #3: None

PLOS authors have the option to publish the peer review history of their article (what does this mean?). If published, this will include your full peer review and any attached files.

Reviewer #1: No

Reviewer #3: No

Figure Files:

Data Requirements:

Reproducibility:

References:

---

## [Decision Letter · Decision Letter 2]

6 Feb 2023

Dear Holman,

We are pleased to inform you that your manuscript 'A behavioral and modeling study of control algorithms underlying the translational optomotor response in larval zebrafish with implications for neural circuit function' has been provisionally accepted for publication in PLOS Computational Biology.

Best regards,

Joseph Ayers, PhD

Academic Editor

PLOS Computational Biology

Wolfgang Einhäuser

Section Editor

PLOS Computational Biology

Reviewer's Responses to Questions

**Comments to the Authors:**

Reviewer #1: The authors have addressed all my concerns thoroughly. The sections on spatial frequency tuning and OMR ratio/over-compensation are nice now and will be helpful to experimenters continuing this type of work.

Reviewer #3: No further comments as earlier issues were addressed.

**Have the authors made all data and (if applicable) computational code underlying the findings in their manuscript fully available?**

Reviewer #1: Yes

Reviewer #3: None

PLOS authors have the option to publish the peer review history of their article (what does this mean?). If published, this will include your full peer review and any attached files.

Reviewer #1: No

Reviewer #3: No

---

## [Editor Report · Acceptance letter]

20 Feb 2023

PCOMPBIOL-D-22-00397R2 

A behavioral and modeling study of control algorithms underlying the translational optomotor response in larval zebrafish with implications for neural circuit function

Dear Dr Holman,

I am pleased to inform you that your manuscript has been formally accepted for publication in PLOS Computational Biology. Your manuscript is now with our production department and you will be notified of the publication date in due course.

With kind regards,

Zsofi Zombor
